# The compression-error trade-off for large gridded datasets

Jeremy D. Silver[1] and Charles S. Zender[2]

[1]School of Earth Sciences, University of Melbourne, Australia
[2]Departments of Earth System Science and of Computer Science, University of California, Irvine, USA
*Correspondence to:* J. D. Silver (jeremy.silver@unimelb.edu.au)

**Abstract.** The netCDF-4 format is widely used for large gridded scientific datasets, and includes several compression methods: lossy linear scaling and the non-lossy deflate and shuffle algorithms. Many multidimensional geoscientific datasets exhibit considerable variation over one or several spatial dimensions (e.g. vertically) with less variation in the remaining dimensions (e.g. horizontally). On such datasets, linear scaling with a single pair of scale and offset parameters often entails considerable loss of precision. We introduce an alternative compression method called "layer-packing" that simultaneously exploits lossy linear scaling and lossless compression. Layer-packing stores arrays (instead of a scalar pair) of scale and offset parameters. An implementation of this method is compared with lossless compression, storing data at fixed relative precision (bit-grooming) and scalar linear packing in terms of compression ratio, accuracy and speed.

When viewed as a trade-off between compression and error, layer-packing yields similar results to bit-grooming (storing between 3 and 4 significant figures). Bit grooming and layer-packing offer significantly better control of precision than scalar linear packing. Relative performance, in terms of compression and errors, of bit-groomed and layer-packed data were strongly predicted by the entropy of the exponent array, and lossless compression was well predicted by entropy of the original data array. Layer-packed data files must be "unpacked" to be readily usable. The compression and precision characteristics make layer-packing a competitive archive format for many scientific datasets.

**Keywords.** netCDF-4; HDF5; Lossy compression; Data storage format

## 1 Introduction

The volume of both computational and observational geophysical data has grown dramatically in recent decades, and this trend is likely to continue. While disk storage costs have fallen, constraints remain on data storage. Hence, in practice, compression of large datasets continues to be important for efficient use of storage resources.

Two important sources of large volumes of data are computational modelling and remote-sensing (principally from satellites). These data often have a "hypercube" structure, and are stored in formats such as HDF5 (http://www.hdfgroup.org), netCDF-4 (http://www.unidata.ucar.edu/netcdf) and GRIB2 (http://rda.ucar.edu/docs/formats/grib2/grib2doc/). Each of these has their own built-in compression techniques, allowing data to be stored in a compressed format, while simultaneously allowing access to the data (i.e. incorporating the compression/decompression algorithms into the format's API). These compression methods are either "lossless" (i.e. no precision is lost) or "lossy" (i.e. some accuracy is lost).

In this study, we examine the advantages and trade-offs of in allowing for different treatment of dimensions in the compression process. One motivation for this work is improving the lossy compression ratios typically achieved with HDF5/netCDF4, so they are more comparable with impressive compression achieved by GRIB2. Records in GRIB2 are strictly two-dimensional and this format allows for only a limited set of predefined metadata. However GRIB2 offers excellent compression efficiency (Caron, 2014), built upon the JPEG image compression methods; it is therefore well-suited for operational environments. By contrast, HDF5/netCDF4 provide a highly flexible framework, allowing for attributes, groups, and user-defined data types. This format is well-suited for more experimental environments and allows introduction of user-defined parameters to guide and improve compression. We use this flexibility to develop a lossy compression algorithm, termed "layer-packing", which combines desirable features from both GRIB2 and HDF5/netCDF4.

Layer-packing is a variant of compression via linear scaling that exploits the clustering of data values along dimensional axes. We contrast this with other compression methods (rounding to fixed precision, and simple linear scaling) that are readily available within the netCDF4/HDF5 framework. The performance is quantified in terms of the resultant loss of precision and compression ratio. We also examine various statistical properties of datasets that are predictive for their overall compressibility and the relative performance of layer-packing or rounding to fixed precision.

## 2   Methods

This section outlines the implementation of the layer-packing and -unpacking methods, the test data sets, evaluation metrics and the performance of the compression methods on the test cases considered.

### 2.1   Compression algorithms

#### 2.1.1   Deflate and shuffle

The "deflate" algorithm (Deutsch, 2008) is a widely-used, lossless compression method, available within most modern installations of the netCDF4 or HDF5 API. The deflate algorithm breaks the data into blocks, which are compressed via the LZ77 algorithm (Ziv and Lempel, 1977, 1978) combined with Huffman encoding (Huffman, 1952). The LZ77 algorithm searches for duplicated patterns of bytes within the block. The Huffman coding step involves substituting frequently used "symbols" with shorter bit-sequence and rarer symbols with longer bit-sequence.

When used in the netCDF4/HDF5 framework, the deflate algorithm is often applied together with the shuffle filter (HDF5 Group, 2002). The shuffle filter does not compress data as such, but instead changes the byte ordering of the data; the first byte of each value is stored in the first chunk of the data stream, the second byte in the second chunk, and so on. This tends to improve the compressibility of the data, particularly if there is autocorrelation in the data stream (i.e. data that are close in value tend to appear close together).

In all of the following work, we have used the deflate algorithm and the shuffle filter together, and are henceforth referred to collectively as DEFLATE for brevity. In the results presented below, compression via DEFLATE was performed with the

`ncks` tool of the NCO bundle (Zender, 2008). The compression level (an integer, taking values between 1 and 9, where 1 means fastest and 9 means greatest compression) dictates the amount of searching for duplicated patterns that is performed within the LZ77 step. For all applications (either as an independent method or in combination with the methods outlined below), the same compression level was used within DEFLATE (namely, level 4).

When using the netCDF4 framework to compress a variable, the user must define the size of the "chunk" within which compression occurs. In all the results presented here (both for DEFLATE and for the other compression methods), the chunk size was equal to the layers packed using layer-packing (see Section 2.1.3). The same analyses were performed setting the chunk sizes to encompass the whole variable, and the conclusions reached in this article were essentially the same.

### 2.1.2   Scalar linear packing

One can compress a field using a lower-precision value (e.g. as two-byte integers rather than four-byte floating point numbers) and a linear transformation between the original and compressed representations; this process is termed "packing". In the common case of representing four-byte floats as two-byte integers, there is already a saving of 50% relative to its original representation (ignoring the small overhead due to storing parameters involved in the transformation). We call this "scalar linear packing" (or LIN, for short) and it is the standard method of packing in the geoscience community. Its attributes are defined in

the netCDF User Guide (Unidata, 2016) and it has a long and wide tradition of support, including automatic interpretation in a range of netCDF readers.

    Scalar linear packing to convert floating-point data to an unsigned two-byte integer representation is outlined below:

$maxPackedValue = 2^{16} - 1 = 65535$

$vMin$ = minimum value of $data$

$vMax$ = maximum value of $data$

$add\_offset = vMin$

$scale\_factor = (vMax - vMin)/maxPackedValue$

$packed = \text{uint16}((data - add\_offset) / scale\_factor )$

where $vMin$, $vMax$, $add\_offset$ and $scale\_factor$ are scalar floating point values, $data$ is a floating point array and $packed$ is

25 an unsigned two-byte integer array of the same dimension as $data$, and the function $\text{uint16}(\cdot)$ converts from floating point to unsigned two-byte integer. Care must be taken in handling infinite, not-a-number or undefined values (such details are omitted here). The value of $maxPackedValue$ listed uses all values in the two-byte integer range to represent floats; one can choose a different value of of $maxPackedValue$, leaving some two-byte values for the special floating-point values. The values of $add\_offset$ and $scale\_factor$ must be stored along with $packed$ to enable the reverse transformation:

$unpacked = \text{float}(packed) \times scale\_factor + add\_offset$

    Data packed with this method often can be compressed substantially more than the 50% noted above. This is done by applying DEFLATE to the packed data; this was done for all datasets compressed with LIN here. In this study, compression via DEFLATE was performed with the `ncpdq` tool of the NCO bundle (Zender, 2008).

### 2.1.3 Layer-packing

In many applications in the geophysical sciences, a multidimensional gridded variable varies dramatically across one dimension while exhibiting a limited range of within slices of this variable. Examples include variation in atmospheric density and water vapour mixing ratio with respect to height, variation in ocean temperature and current velocity with respect to depth, and
variation in atmospheric concentrations of nitrogen dioxide with respect to height and time.

We will use the term "thick dimensions" to denote those dimensions that account for the majority of the variation in such variables, "thin dimensions" to denote the remaining dimensions; in the case of the first example above (assuming a global grid and a geographic coordinate system), the vertical dimension (pressure or height) is thick, and the horizontal dimensions (latitude, longitude) and time are thin. We will use the term "thin slice" to describe a slice through the hypercube for fixed
values of the thick dimensions. Note that there are cases with multiple thick dimensions, such as the third example noted above.

Scalar linear packing applied to such gridded variables will result in a considerable loss of accuracy. This is because in order to cover the scale of variation spanned by the thick dimensions, the range of the individual thin slices will be limited to a subset of available discrete values. To reduce loss of precision within the thin slice, one can store for each variable *arrays*
of scale-offset pairs, with size corresponding to the thick dimensions only. This is the key innovation of the layer-packing technique (or LAY for brevity). Layer-packing, as discussed here, was implemented via the following algorithm (described here in pseudocode):

    **for** *var* in *vars* **do**

       **if** (*var* in *splitVars*) and intersection(dimensions[*var*],*splitDims*) is not None **then**

*theseSplitDims* = intersection(dimensions[*var*],*splitDims*)

          *theseDimLens* = lengths of *theseSplitDims*

          *iSplitDim* = indices of *theseSplitDims* in dimensions[*var*]

          *splitDimIdxs* = all possible index combinations for *theseSplitDims*

          *nSplits* = number of combinations given in *splitDimIdxs*

*nDim* = length of dimensions[*var*]

          *indices* = list of length *nDim*, with each element equal to Ellipsis

          *data* = the data array

          *packed* = array of zeros, type is uint16, with the same shape as *data*

          *add_offset* = array of zeros, type is float, with dimensions given by *theseDimLens*

*scale_factor* = same as *add_offset*

          **for** *iSplit* in range(0,*nSplits*) **do**

             *indices*[*iSplitDim*] = *splitDimIdxs*[*iSplit*]

             *thinSlice* = *data*[*indices*]

             *vMin* = minimum value of *thinSlice*

$vMax$ = maximum value of *thinSlice*

$add\_offset[splitDimIdxs[iSplit]] = vMin$

$scale\_factor[splitDimIdxs[iSplit]] = (vMax - vMin)/maxPackedValue$

$packed[indices] = \text{uint16}(\ (thinSlice - add\_offset)\ /\ scale\_factor\ )$

5      **end for**

       write *packed*, *add_offset*, *scale_factor* to file

    **else**

       write the *data* array to file

    **end if**

10    **end for**

where *vars* are the variables in the file, *splitVars* are those variables that should be layer-packed, and *splitDims* are the thick dimensions for layer-packing.

The two-byte representation halves the storage cost of the array itself, however arrays of scale factors and linear offsets must also be stored and this adds to the total space required (n.b. these are often negligible relative to the size of the packed data array). Compression is generally significantly improved by applying DEFLATE and this was done for all datasets presented here.

Further details about the implementation and the storage format are given in the Supplementary Material document. The code to perform layer-packing described in this article was written as stand-alone command-line tools in Python (v. 2.7.6). These are freely available on https://github.com/JeremySilver/layerpack. Beyond the standard Python installation, it requires that the `numpy` and `netCDF4` modules are installed.

### 2.1.4   Bit grooming

One can store the data at a fixed precision (i.e. a chosen number of significant digits, or NSD). This method is known as "bit-grooming" and is detailed by Zender (2016) and implemented in the NCO package (Zender, 2008). If bit-groomed data remain uncompressed in floating-point format this coarsening will not affect the file size, however the application of the deflate/shuffle algorithms will in general improve the compressibility of the coarsened data, as they will be represented by a smaller number of discrete values. For further explanation, we must briefly summarize how floating-point numbers are represented by computers.

Single-precision floating-point numbers occupy 32 bits within memory. Floating-point numbers are represented as the product of a sign, significand and an exponential term:

$$\text{datum} = \text{sign} \cdot \text{significand} \cdot \text{base}^{\text{exponent}}$$

The IEEE standard specifies that the sign accounts for 1 bit, the significand (also known as the mantissa) 23 bits and the exponent 8 bits. The base is 2 by convention. The sign is an integer from the set $\{-1, 1\}$, the significand is a real number in the range $[0.0, 1.0)$ and the exponent is an integer between -128 and +127; see, for example, Goldberg (1991) for further details.

Bit grooming quantizes[1] data to a fixed number of significant digits (NSD) using bitmasks, not floating-point arithmetic. The NSD bitmasks alter the IEEE floating point mantissa by changing to 1 (bit setting) or 0 (bit shaving) the least significant bits that are superfluous to the specified precision. Sequential values of the data are alternately shaved and set, which nearly eliminates any mean bias due to quantization (Zender, 2016). To guarantee preserving 1–6 digits of precision, bit grooming retains $5, 8, 11, 15, 18$ and $21$ explicit mantissa bits, respectively, and retains all exponent bits.

In the following we compared storing 2, 3, 4 and 5 significant digits; these are denoted NSD2, NSD3, NSD4 and NSD5, respectively. Similar to LIN and LAY, DEFLATE was applied together with rounding. In this study, compression via bit-grooming was performed using the `ncks` tool within the NCO package (Zender, 2008).

## 2.2 Datasets

In the following tests, we compared a total of 255 variables from six datasets. Each variable was extracted individually to file as uncompressed netCDF, and the file was then compressed using the methods described, allowing for computation of compression and error metrics described below. The datasets are summarised in Table 1. Further details are provided in the online Supplementary Material section.

The variables chosen from these datasets were those with the largest number of data-points overall. For example in datasets 2-5, variables without a vertical coordinate were not considered in the analysis, since these account for only a small fraction of the total data. A small number of the variables that would otherwise be included (based on the dimensions alone) were excluded due to the occurrence in seemingly random data (i.e. of all magnitudes) in regions of the array where values were not defined (in the sense of sea-surface temperatures over land points), which we believe should have been masked with a fill-value; the rationale for excluding these variables is first, that these regions did not appear to contain meaningful data and that the extreme range of the seemingly-random data led to gross outliers in the distribution of error statistics for LIN and LAY in particular.

## 2.3 Error and compression metrics

The methods are compared with two metrics. The first relates to the compression efficiency. Compression ratios are defined as (uncompressed size)/(compressed size), and as such larger values indicate greater compression. The second metric relates to the accuracy (or, seen another way, the error) of the compressed data relative to the original data. The error of DEFLATE is zero since it ensures lossless compression. The remaining methods cause some loss of precision.

Error was quantified by the root mean-square difference between the original and the compressed variables. However in order to compare results across variables with different scales and units, the errors must be normalized somehow. We considered four different normalization methods, which emphasized different aspects of the error profile. The errors were normalized either by the standard deviation or the mean of the original data, and these were either calculated separately per thin slice or across the entire variable – the rationale is as follows.

---

[1] The process of quantization means mapping, in this case via a process similar to rounding, from a large set of possible inputs (in this case the full set of real numbers representable as floating-point values) to a smaller set (in this case those floating-point values defined to a chosen NSD).

| ID | Description | Grid type | Dims | TS Dims | # vars |
|----|-------------|-----------|------|---------|--------|
| 1 | Global 3-D NWP reanalyses | Rectangular | $(n_x, n_y, n_z, n_t) = 240 \times 121 \times 37 \times 16$ | $(n_x, n_y) = 240 \times 121$ | 14 |
| 2 | 3-D CTM output | Rectangular | $(n_x, n_y, n_z, n_t) = 9 \times 10 \times 56 \times 172$ | $(n_x, n_y) = 9 \times 10$ | 77 |
| 3 | 3-D NWP model output | Rectangular | $(n_x, n_y, n_z) = 165 \times 140 \times 32$ | $(n_x, n_y) = 165 \times 140$ | 20 |
| 4 | Global 3-D NWP reanalyses | Rectangular | $(n_x, n_y, n_z, n_t) = 288 \times 144 \times 42 \times 8$ | $(n_x, n_y) = 288 \times 144$ | 11 |
| 5 | Dust transport-dispersion model | Rectangular | $(n_x, n_y, n_z) = 192 \times 94 \times 28$ | $(n_x, n_y) = 192 \times 94$ | 15 |
| 6 | 3-D coupled NWP-CTM output | Irregular | $(n_{x'}, n_z) = 48602 \times 30$ | $n_{x'} = 48602$ | 118 |

**Table 1.** Summary of the datasets used in this study. Abbreviations: ID = index, NWP = numerical weather prediction, CTM = chemistry-transport model, Dims = Dimensions of the variable, TS Dims = Dimensions of the thin slice, # vars = number of variables per dataset. The dimension sizes are indicated as: $n_x$ = length of the east-west dimension, $n_y$ = length of the north-south dimension, $n_z$ = length of the vertical dimension, $n_t$ = length of the time dimension and $n_{x'}$ = length of the generalized horizontal coordinate dimension (used for the unstructured grid in last dataset only).

When normalizing by the mean (of the entire variable, or of the thin-slice), variables with a low mean-to-standard-deviation ratio (e.g. potential vorticity) will show larger errors using the layer-packing compression. Whereas when normalizing by the standard deviation, variables with a high mean-to-standard-deviation ratio will show larger errors using the bit-grooming compression (e.g. atmospheric temperatures stored with units of K, concentrations of well-mixed atmospheric trace gases such as $CO_2$ or $CH_4$).

If we calculate the ratio of the RMSE to normalization factor (i.e. the mean or standard deviation) per thin slice and then average across the normalized errors across slices, the resulting metric will be more sensitive to large relative errors within subsections of the data array. The alternative is to calculate the normalization factor across the whole variable, and the resulting metric will be more reflective of relative errors across the entire data array. This may be understood in the context of a hypothetical thee-dimensional array, with values ranging from $O(10^4)$ to $O(10^0)$ across the vertical dimension and a mean value of $O(10^3)$, and errors roughly uniform of $O(10^{-1})$; if relative errors (normalizing by the mean) are calculated for the whole array then they will be $O(10^{-4})$, whereas if calculated across each thin slice separately they will range from $O(10^{-1})$ to $O(10^{-5})$, and may have a mean of $O(10^{-2})$. The case of uniform errors is most likely to arise for LIN, whereas bit-grooming guarantees precision for each individual datum and layer-packing for each thin slice.

## 2.4 Complexity statistics

In order to make sense of which variables compress well or poorly with different methods, a range of statistics were calculated for each variable. Most of these statistics were calculated over two-dimensional hyperslices of the original data arrays and then the value for an individual variable was taken as the average over these hyperslices. A full list of the statistics calculated is given in the Supplementary Material document.

The two most informative statistics that arose from this analysis were based on the entropy of either the original data field or the corresponding exponent field (i.e. decomposing the data array into significand and exponent, and then calculating the entropy of the exponent array). The entropy is a measure of statistical dispersion, based on the frequency with which each value appears in each dataset. Let us denote as $P(x_i)$ the proportion of the array occupied by each unique value $x_i$. For an array containing discrete values $X = \{x_i\}_{i=1}^k$ for $k$ discrete values, the entropy was defined as

$$H(X) = \mathbb{E}[-\log_2(P(X))] = -\sum_{i=1}^{k} P(x_i) \log_2(P(x_i)) \tag{1}$$

For an array of $K$, the entropy has a maximum value of $\log_2(K)$, which will arise if all values are unique. For single-precision arrays of size $2^{32} \approx 4.29 \times 10^9$ or larger, the maximum entropy is equal to 32.

In order to normalize for these limitations to the entropy of a finite dataset, for each case the entropy was normalized by the maximum theoretical value attainable for that dataset, which was taken to be $\log_2(K')$, where $K'$ is the number of elements in the thin slice (in each case $K \ll 2^{32}$). In the case of the entropy of the exponent array, the normalization was based on the $\min(\log_2(K'), 8)$, since the maximum entropy of an 8-bit field (recall, 8 bits are used for the exponent of a floating point) is 8.

It was found that some of the datasets compressed significantly using DEFLATE only. This was often due to a high proportion of zero or "missing" values. Variables were classified as either "sparse" (highly compressible or otherwise relatively simple) or "dense" (all other variables). Sparse variables were chosen to be those satisfying any one of the following conditions: the compression ratio is greater than 5.0 using DEFLATE, the fraction of values equal to the most common value in the entire variable is greater than 0.2, and the fraction of hyperslices where all values were identical is great than 0.2. These definitions were somewhat arbitrary and other classifications may be preferable, but it is seen (e.g. in Figures 1C, 3A and 3B) that sparse variables do not always follow the same pattern as dense variables. Of the 255 variables in total, 181 were classified as dense. The breakdown among the different categories is given in Table 1 in the Supplementary Material document.

## 2.5 Compression and error results

Figure 1A shows the distribution of compression ratios, normalized errors and timing statistics for the different methods. In the case of the compression ratios and normalized errors, results are presented separately for the dense and sparse variables. For dense variables, the median compression ratios were 1.3 (DEFLATE), 3.2 (NSD2), 2.4 (NSD3), 2.0 (NSD4), 1.6 (NSD5), 4.2 (LIN) and 2.6 (LAY). For sparse variables, the median compression ratios were 2.0 (DEFLATE), 4.3 (NSD2), 3.3 (NSD3), 2.8 (NSD4), 2.3 (NSD5), 7.4 (LIN) and 5.2 (LAY). It can be seen that LIN gave the greatest compression, and the LAY compression ratios were comparable with those of NSD2 or NSD3.

The median compression times (Figure 1B) normalized relative to the DEFLATE compression time were 0.82 (NSD2), 0.91 (NSD3), 0.79 (NSD4), 0.91 (NSD5), 0.57 (LIN), 3.45 (LAY compression) and 1.84 (LAY extraction). Differences between the bit-grooming methods were relatively small and slightly faster than DEFLATE alone, whereas the LIN compression was nearly twice as fast as DEFLATE. These values are consistent with DEFLATE compressing twice as much data for bit-grooming as for LIN, which store four and two bytes per value, respectively. The LAY times (both compression and decompression) were significantly slower than for the other methods, particularly for compression; this is most likely due to differences in

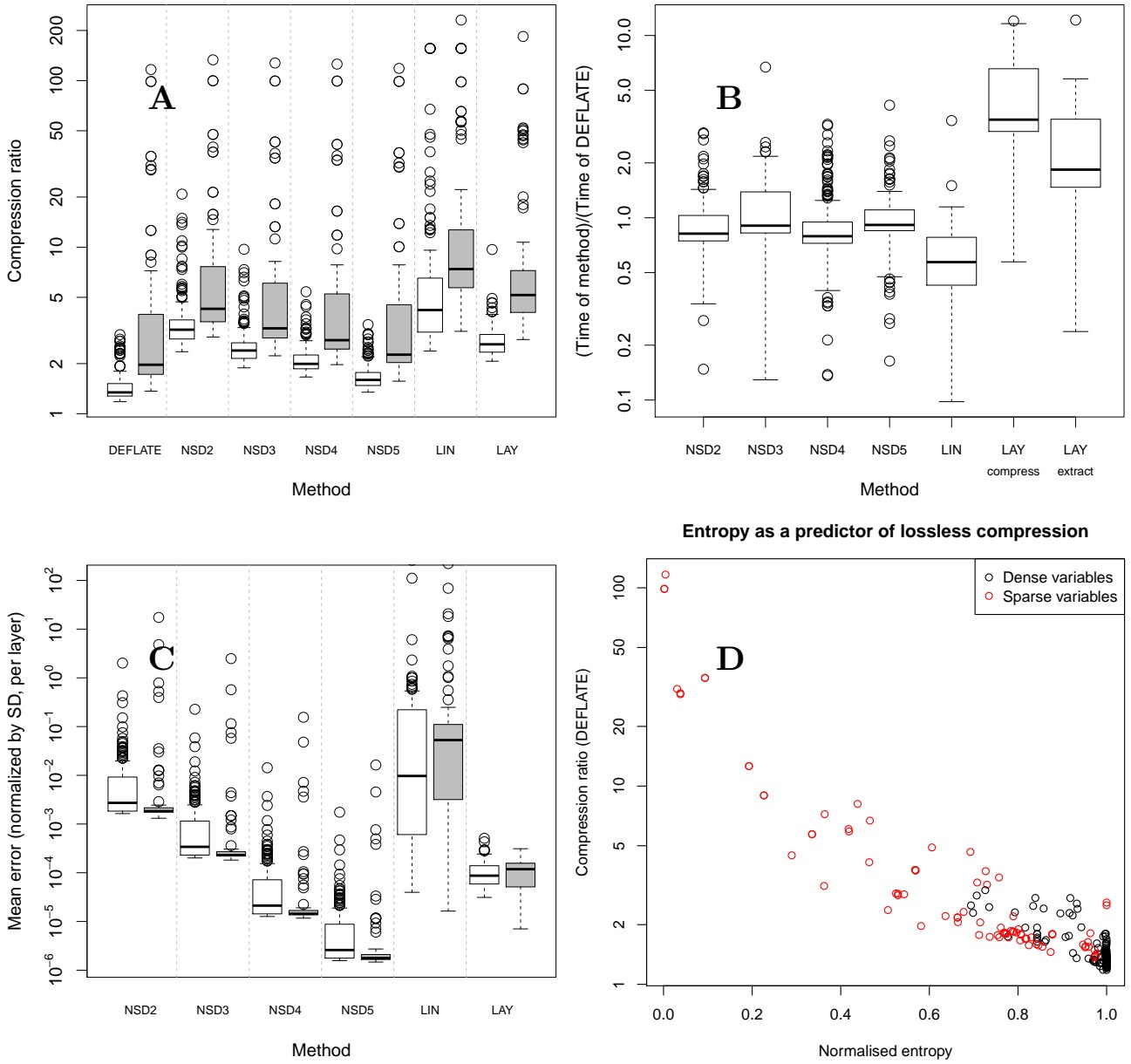

**Figure 1. A**: Distribution of compression ratios (original file size divided by compressed file size) measured for seven methods, applied to six test datasets (higher is better), plotted separately by dense variables and sparse variables (white and grey boxes, respectively). The box-plots (in panels A, B and C) were defined as follows: the thick line and the center of the box denotes the median, the bottom and top of the box show $q_{0.25}$ and $q_{0.75}$ (respectively) or the 0.25 and 0.75 quantiles of the distribution, the whiskers extend from $q_{0.25} - 1.5 \cdot \mathrm{IQR}$ to $q_{0.25} + 1.5 \cdot \mathrm{IQR}$, where IQR is the inter-quartile range ($q_{0.75} - q_{0.25}$) and the points shown are all outliers beyond this range. **B**: Distribution of scaled compression/decompression times for each method (lower is better), with LAY represented twice (for compression and extraction). These times are normalized by the compression time from DEFLATE. **C**: Distribution of errors, normalized by the per-layer standard deviation. **D**: The achieved lossless compression ratios (i.e. from DEFLATE) as a function of the normalized entropy (1.0 corresponding to the maximum theoretical for the data).

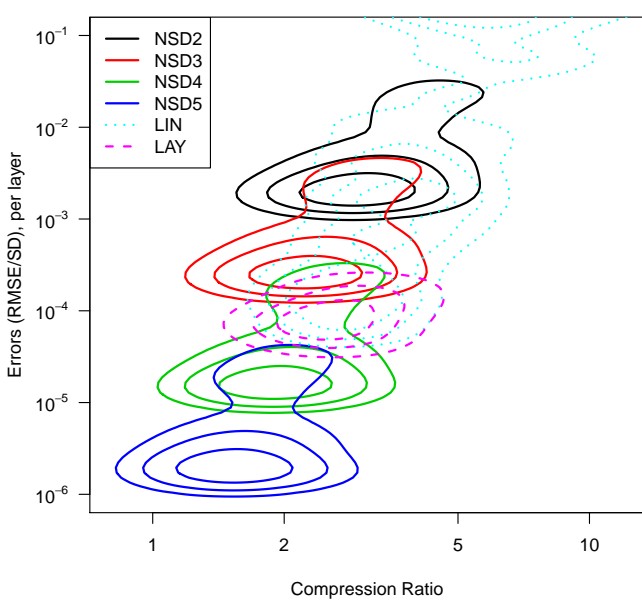

**Figure 2.** The relationship between normalized errors and compression ratio for the lossy compression methods considered. The three contours for each method show the bounds within which the two-dimensional kernel-smoothed distribution integrates to 0.25, 0.5 and 0.75, respectively. Only dense variables were used to produce this plot. The normalized errors (the $y$-axis coordinate) were measured per layer and then averaged.

implementation, as this implementation of LAY was programmed in Python while the other methods used compiled C/C++ utilities. We believe that the overhead from loading some of the python libraries used in the implementation of LAY may cause this method to be relatively slower for smaller files; we note that most of the variables considered are relatively small, with uncompressed file sizes ranging from 1.9 MB to 65.6 MB. This hypothesis was supported by a test case where the suite of

5    compression methods were applied to a much larger array[2] of size 1.5GB, the compression times were 103 s (DEFLATE), 89 s (NSD2), 107 s (NSD3), 97 s (NSD4), 109 s (NSD5), 69 s (LIN) and 70 s (LAY), while the unpacking time for LAY was 133 s.

For all methods considered except DEFLATE, the compression comes at the expense of precision; the distribution of resultant errors is shown in Figure 1C (and Figure 1 of the online Supplementary Material). For dense variables, the median relative errors shown in Figure 1C are $1.8 \cdot 10^{-3}$ (NSD2), $2.3 \cdot 10^{-4}$ (NSD3), $1.4 \cdot 10^{-5}$ (NSD4), $1.8 \cdot 10^{-6}$ (NSD5), $5.3 \cdot 10^{-2}$ (LIN),

10    $1.2 \cdot 10^{-4}$ (LAY). Unsurprisingly, with each additional significant digit of precision requested of bit-grooming, the normalized errors fall by a factor of 10. The errors of LAY are comparable to those of NSD3 or NSD4 while for the metric shown in Figure 1C (RMSE/SD, calculated separately per thin slice, then averaging the ratios), LIN displays much larger errors than the

[2]The ERA-Interim (Dee et al., 2011) east-west wind component at 241 latitude, 480 longitudes, 60 vertical levels and 124 times, spanning 2015-07-01 00:00 UTC to 2015-07-31 18:00 UTC at 6-hourly intervals, converted from its original GRIB format. This dataset can be accessed through the ECMWF's public dataset portal (http://apps.ecmwf.int/datasets/)

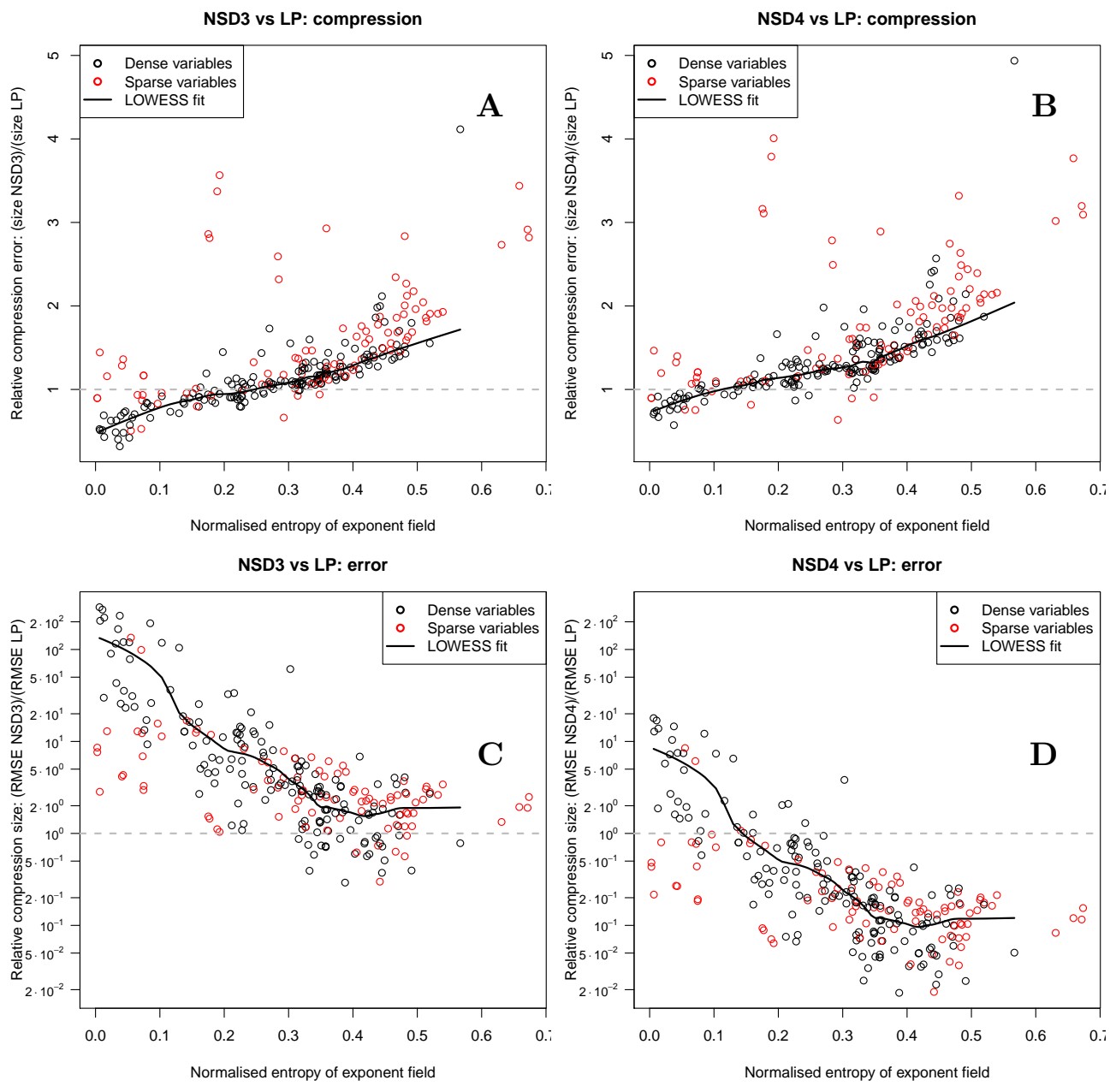

**Figure 3.** Relative performance of LAY compared with NSD3 (left column) or NSD4 (right column), in terms of compression (top row) and errors (bottom row). The grey dashed line indicates values of 1.0 on the $y$-axis. The LOWESS fit was based on the dense variables alone.

other methods – the errors for LIN are, more than for the other lossy compression methods in this comparison, sensitive to the error metric used and this is discussed below. Dense variables show some differences in the distribution of errors compared to sparse variables; errors normalized by the standard deviation appear smaller and the errors normalized by the mean appear larger. This is because many of the sparse variables are zero at most points, and in many cases this tends to reduce the mean and increase the standard deviation (compared to examining only the non-zero values).

The choice of error metric is ambiguous and leads to slightly different different results. Four different normalization factors for the RMSE were considered (shown in Figure 1 of the online Supplementary Material). It can be seen that the comments about the bit-grooming and LAY methods in the preceding paragraph hold regardless of the normalization method, whereas LIN shows much higher standarized errors if errors are normalized within thin-slices and then averaged (due to the reasons explained in the last part of Section 2.3). Bit-grooming keeps precision loss to known bounds for each individual datum, LAY leads to roughly constant errors within thin-slices, and LIN results in roughly constant errors for the whole variable. This last point is illustrated in Figure 4 (especially panels A, B, C and E) of the Supplementary Material, which shows horizontal profiles of the RMSE, mean and standard deviation for a sample of six variables from among the 255 considered; Figure 5 in the Supplementary Material shows the corresponding relative errors.

The pairwise relationship between compression and error is shown in Figure 2, with the distribution shown based only on dense variables; the error metric used in this figure is the average (across layers) of the RMSE normalized by the per-layer standard deviation, which tends to highlight large relative errors in certain sections of the data array (most notably for LIN). We see that the bit-grooming and LAY methods form something of a continuum, with LAY falling between NSD3 and NSD4. The linear slope on a log-log plot is suggestive of a power-law relationship, which would be consistent with fundamental constraints on compression potential consistent with rate-distortion theory (Berger, 2003).

The question of when LAY is preferable to NSD3 or NSD4 can be addressed with reference to the complexity statistics. Among those complexity metrics considered, the normalized entropy of the data field proved to be the best predictor of compression in the lossless case (DEFLATE). By contrast, the best predictor of the relative performance of LAY to the two bit-grooming methods was the normalized entropy of the exponent field (NEEF). By "best predictor", we mean that these were, respectively, the most strongly correlated among the metrics considered with the DEFLATE compression ratios and the relative error or compressed file size. In the case of lossless compression, the correlation of the log of the compression ratio with the normalized entropy was over 0.9 (the next highest correlation was below 0.8; all variables were included), while for differentiating bit-grooming and LAY, the absolute correlations between the NEEF and the log of the file size ratio or the log of the RMSE ratio as shown in Figure 3 were both over 0.8 (c.f. the next highest correlations were around 0.6; only dense variables were included).

The trade-off between error and compression is evident: as the NEEF increases, the bit-groomed file-sizes become larger than the corresponding LAY file-sizes, while the errors of resulting from LAY grow relative to those of bit-grooming (Figure 3). The LAY file sizes were larger than those of NSD3 or NSD4 when the NEEF greater than 0.25 or 0.1, respectively (Figure 3, upper row). Errors of LAY were generally less than those of NSD3, while the errors for NSD4 were smaller compared to LAY for values of the NEEF greater than 0.15 (Figure 3, lower row). In most cases the choice between LAY and NSD3 or NSD4

implies a choice between smaller file sizes or smaller errors (an exception being when then NEEF is greater than 0.25, and LAY yields both smaller files and lower errors than NSD3).

The reason that the NEEF differentiates the relative performance between bit-grooming and LAY can be understood by the nature of the errors induced by the two techniques. Bit-grooming guarantees constant *relative* errors for each individual datum, whereas LAY results in errors that are roughly constant in *absolute* magnitude across a thin-slice. Assuming that the variable is dense, if the data within each thin-slice vary little (corresponding to a low NEEF), then LAY will achieve relatively small errors and since the compressed field contains more information, the resultant file size will be larger.

One interesting aspect of these results is that the entropy is defined *independent* of the values in the distribution (Figure 1), based solely on the frequencies of values in the array, yet the NEEF proved to be more informative regarding relative performance than other metrics that accounted for range in magnitude (e.g. the range or standard deviation of the exponent field, the logarithm of the largest non-zero value divided by the smallest non-zero value). This is likely due to the fact that the RMSE summarises the error distribution, weighting both by size and frequency, and the entropy is a measure of statistical dispersion.

Given the clear relationship between the normalized entropy of the data field (NEDF) and the compression ratios achieved by DEFLATE alone (Figure 1D), it leads to the question of whether the NEDF is predictive of compression ratios achieved by the lossy methods. The NEDF is indeed highly predictive of the compression ratio of the reduced-precision fields as well (Figure 2 of the Supplementary Material), with the absolute correlation exceeding 0.85 in each case. This may be expected, given the already clear relationship between these two parameters. Furthermore, the reduction in the NEDF is seen to be predictive of the "compression improvement", which we define as the DEFLATE-compressed file-size divided by the file-size achieved by the lossy compression method; this relationship is most apparent for those methods with the highest compression ratios, namely LIN, LAY and NSD2 (see Figure 3 of the Supplementary Material). The linear relationship, however, offers only a partial explanation (the correlation is over 0.8 for LIN and LAY, and over 0.6 for NSD2 and NSD3), however it appears to explain most of the variation when the lossy method achieves two-fold or greater compression relative to that obtained by DEFLATE.

## 3  Discussion

Layer-packing was compared with scalar linear packing, bit-grooming and lossless compression via the DEFLATE algorithm. The lossy methods form a continuum when one compares the resultant compression ratios and normalized errors. The trade-off between error and compression has been shown elsewhere (e.g. Berger, 2003). Despite the fact that LAY and LIN represented the data as two-byte integer arrays, which occupy half the storage of four-byte floating point numbers (ignoring the relatively minor contribution to LAY from the much smaller accompanying scale and offset arrays), it can be seen that both methods effectively fit into the continuum spanned by bit-grooming, which represents data as four-byte floats (Figure 2).

In this study, we have effectively separated the precision-reduction from the compression itself. This is because the bit-grooming, LAY and LIN methods can be thought of as preconditioners for the same lossless compression algorithm (namely DEFLATE). The reduction in entropy due to these preconditioners explains a large part of the improved compression above

lossless compression. This concept could be extended to develop methods for automatically determining the right precision to retain in a dataset.

This study did not extend to the comparison with other lossless filters for compression of the precision-reduced fields, nor did it compare the results with other lossy compression techniques. While it is possible that the findings presented here extend beyond the deflate and shuffle technique, other lossy and non-lossy compression algorithms operate in fundamentally different ways. Such an extension to this study may be considered in future, for example, by taking advantage of the fact that the HDF5 API allows for a range of alternative compression filters to be loaded dynamically (HDF5 Group, 2016) to provide faster or more efficient compression than the default algorithms.

A number of such compression filters (both lossy and lossless) have been developed specifically for geophysical datasets, which have different properties compared to plain text, for example. They tend to be multi-dimensional, stored as floating-point numbers and in many cases are relatively smooth (Hübbe et al., 2013). Specialised compression algorithms exist for such datasets that rely on the smoothness properties to confer a certain degree of predictability, which reduces the number of effective degrees of freedom in the dataset. For example, Hübbe et al. (2013) present a lossless algorithm termed MAFISC (Multidimensional Adaptive Filtering Improved Scientific data Compression) that incorporates a series of filters (some of which are adaptive), which in the cases presented gave stronger compression than the other lossless algorithms under consideration. In a similar spirit, Di and Cappello (2016) use a lossy, adaptive, curve-fitting technique that gives highly competitive compression performance while simultaneously constraining relative and/or absolute errors (as defined by the user).

Other authors have shown impressive data-compression rates using methods originally developed for image processing. For example, the GRIB2 format allows for compression using the JPEG-2000 algorithm and format (based on wavelet transforms) to store numerical fields (Skodras et al., 2001). Woodring et al. (2011) describe a work-flow based around JPEG-2000 compression in order to overcome bandwidth-limited connections while quantifying the ensuing reduction in precision. They demonstrate the same compression-error trade-off as illustrated in this work. Robinson et al. (2016) compare lossy compression via JPEG-2000, PNG (another graphics standard) and three video codecs; the video codecs showed very high relative errors (in the order of 0.1 to 1.0), but also very high compression rates (around 300-fold compression).

A number of studies have assessed a range of compression methods on a variety of datasets (e.g. Baker et al., 2014; Di and Cappello, 2016). They show, amongst other things, that no one method provides the most effective compression for all datasets considered. Also apparent is that lossy methods tend to result in higher compression ratios than lossless techniques, and that methods designed specifically for scientific data (e.g. exploiting smoothness when it is present) are often highly competitive.

Layer-packing achieves compression and error results roughly in between bit-grooming storing three or four significant figures. Layer-packing and bit-grooming control error in different ways, with bit-grooming guaranteeing fixed relative errors for every individual datum while layer-packing results in roughly constant relative errors within the hyper-slice across which the packing is applied.

The idea itself behind layer-packing is not new, and forms the basis of compression within the GRIB format, in which each field is a two-dimensional array and compression is performed on fields individually. In one sense layer-packing is more general than that used in GRIB, in that thin-slices are not restricted to being two-dimensional. Our preliminary results (not

shown) showed that the JPEG2000 algorithm yields greater compression compared to the methods presented here for the same level of error; this echoes the findings of Caron (2014), which describe the efficient compression achieved. However like scalar linear packing, JPEG2000 does not offer clear controls about the resultant errors and thus some experimentation (in setting the number of bits per value) is needed to avoid excessive loss of precision (Woodring et al., 2011). More limiting, perhaps, are the

technical procedures required to convert generic netCDF data into GRIB2 format, including meeting constraints on variable and dimension names. The GRIB format only allows storing meta-data that match predefined tables, and is thus nowhere near as general or self-describing as HDF5 (or its derivatives such as netCDF-4). This, combined with the software requirements beyond netCDF (e.g. the JPEG library) for decompression render GRIB-compression unattractive for general purpose usage. Outside of organisations with strong technical support for modelling operations (e.g. operational weather prediction centres),

its suitability may be limited to special purposes and expert users.

Although the methods described here are used only on netCDF data files, they apply equally to other data formats (e.g. HDF4, HDF5). These results are presented as a proof-of-concept only.

The timing information is presented mainly for completeness and a caveat should be raised. As noted above, the compression via DEFLATE, bit-grooming and LIN were performed using tools from the NCO bundle (written in C and C++), whereas

the LAY compression was implemented in Python. The code is presented as a demonstration of layer-packing rather than production code.

Both the scalar linear packing and the layer-packing use the same representation of the data (i.e. two-byte integers), however large differences in the compression and relative errors were found. This is because the compressibility of a packed field is related to the distribution of values within the scaling range. Across a given thin slice, layer-packing will represent values

using the full range of two-byte integers, whereas scalar linear packing will typically use a smaller portion of that range. This increases the loss of precision resulting from scalar linear packing but also entails greater compression.

When considering which compression method to use for an individual dataset, one needs to consider several factors. First, space constraints and the size of the datasets in question will vary considerably for different applications. Second, the degree of precision required will also depend on the application, and may differ between variables within a dataset. The bit-grooming

(storing at least three significant digits) and layer-packing techniques achieved average normalized errors of 0.05% or better, which in many geoscientific applications is much less than the model or measurement errors. Third, datasets vary considerably in their inherent compressibility, which in the cases considered appeared strongly related to the normalized entropy of the data array. Fourth, how data are stored should also reflect how it will be used (e.g. active use versus archiving). A major disadvantage of the layer-packing as described here is that it is essentially an archive format, and needs to be unpacked by

a custom application before it can be easily interpreted. Scalar linear packing is similarly dependent on unpacking (although many netCDF readers will automatically unpack such data, from two-byte integers to four-byte floating point, by default), whereas bit-grooming requires no additional software. Finally, other issues relating to portability, the availability of libraries and consistency within a community also play a role in determining the most appropriate storage format.

## 4  Conclusions

This paper considers layer-packing, scalar linear packing and bit-grooming as a basis for compressing large gridded datasets. When viewed in terms of the compression-error trade-off, layer-packing was found to fit within the continuum of bit-grooming (i.e. when varying the number of significant digits to store), roughly in between storing three and four significant digits. The relative performance of layer-packing and bit-grooming was strongly related to the normalized entropy of the exponent array, and again highlighted the trade-off between compression and errors. Given the variation in compression and accuracy achieved for the different datasets considered, the results highlight the importance of testing compression methods on a realistic sample of the data.

If space is limited and a large dataset must be stored, then we recommend that the standard deflate and shuffle methods be applied. If this does not save enough space, then careful thought should be given to precisely which variables and which subsets of individual variables will be required in future; it often arises that despite a wealth of model output, only a limited portion will ever be examined. Many tools exist for sub-setting such datasets. Beyond this, if further savings are required and if the data need not be stored in full precision, then the appropriate relative precision for each variable should be selected and applied via bit-grooming. Layer-packing should be considered when choosing a compression technique for specialist archive applications.

## 5  Code availability

The python-based command line utilities used for the layer-packing, unpacking and relative-error analysis are available freely at https://github.com/JeremySilver/layerpack.

## 6  Data availability

Of the datasets used in the tests (listed in Section 2.2), datasets 2–6 are available online at https://figshare.com/projects/Layer_Packing_Tests/14480 along with a brief description of each file. Dataset 1 could not be distributed with the other files due to licensing restrictions but can be accessed through the ECMWF's public dataset portal (http://apps.ecmwf.int/datasets/), using the following set of inputs: stream = synoptic monthly means, vertical levels = pressure levels (all 37 layers), parameters = all 14 variables, dataset = interim_mnth, step = 0, version = 1, time = 00:00:00, 06:00:00, 12:00:00, 18:00:00, date = 20080901 to 20081201, grid = $1.5° \times 1.5°$, type = analysis, class = ERA Interim.

*Author contributions.*  J. D. Silver wrote the layer-packing python software, performed the compression experiments and wrote most of the manuscript. C. S. Zender contributed to the design of the study, provided some of the test datasets used in the experiments and contributed to the text.

## 7 Competing interests

The authors declare that they have no conflict of interest.

*Acknowledgements.* The work of J. D. Silver was funded by the University of Melbourne's McKenzie Postdoctoral Fellowship programme. The work of C. S. Zender was funded by NASA ACCESS NNX12AF48A and NNX14AH55A and by DOE ACME DE-SC0012998. We
5  thank Peter J. Rayner (University of Melbourne) for useful discussions. Three anononymous reviewers provided constructive comments and suggestions on the manuscript.

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
