# Peer review of "The compression-error trade-off for large gridded datasets"

_Geoscientific Model Development, 2016_

## Referee Comment (RC1) · Anonymous Referee #1 · 22 Aug 2016

————————————- Summary: ————————————-

The variation in a gridded dataset may be notably different in each spatial direction. Taking this into account when applying (lossy) compression can improve the resulting precision. They propose a technique called layer packing that achieves better compression ratios than a lossless approach and preserves precision better than a comparable lossy approach.

————————————- General comments: ————————————-

(1) This paper addresses an important issue because data compression is very much needed to mitigate large data volumes in geophysical data.

[Figure]

(2) Treating the dimensions differently when applying lossy compression to gridded data makes a lot of sense.

(3) Section 1 and 2 need some rearranging and improvement (more details are given below in "specific comments") in terms of introducing the ideas and terminology. It could be better to shorten the introduction and then really explain the methods well in section 2.

(4) The audience for this work may not be too familiar with compression techniques other than just using defaults in netcdf, so improving the explanations for the techniques would be helpful. (For example, defining a "deflate and shuffle" algorithm).

(5) The paper's contribution should be clarified in the introduction (section 1). It is not clear to me whether "layer packing" is a new idea that is first presented here. (It is mentioned a bit more clearly in section 3).

(6) For this paper to really impact the broader geophysical data community, I feel that more details on the compression approaches need to be provided.

(7) More details on the datasets are needed to be able to understand why compression effects the each differently. Perhaps look at variables instead of multi-variable datasets?

———————————— Specific comments: ————————————-

(1) page 2, par. 1: For this audience, please give more explanation of the techniques. For example, please provide more explanation of how "deflate and shuffle" works (rather than just pointing to a reference).

(2) page 2, line 22: "Linear packing with a single scale-offset parameter" - is discussed here but not well-defined. Note that "packing" is later defined in line 32. Then "scalar linear packing" on p.3. line 2. In general, the terminology used and defined in this paragraph is hard to follow in that it is sometimes defined after being used. (Also, is "linear packing with a single scale-offset parameter" the same as "scalar linear packing"?)

(3) p.2, line 29: I'm not sure the audience will be familiar with "quantization" (like the audience for a CS publication would).

(4) section 2.1.1 (Layer packing)" Here I would suggest providing more detail (maybe an example) - particularly if this approach is the main contribution of the paper. Rather than providing syntax details, consider defining/explaining the parameters (the reader may not be familiar with what these are) here.

(5) section 2.1.2, line 15: Explain what "level" means in the algorithm.

(6) section 2.1.2, line 17: Explain a shuffle filter.

(7) section 2.3: Regarding the datasets listed, more information about the model source (other than acronym and reference) would be helpful - especially in interpreting the results. Without more details, I cannot really understand how the datasets differ and, therefore, why/how they would respond to compression differently. For example, the number of grid points are given - but does this number represent a domain on the entire globe for all datasets? The number of vertical levels is listed, but do all models simulate to the same height? What is the time dimension? Hourly? Monthly averages? Is the time dimension the same for each data et?

(8) Fig 1: For compression results, I think it would be more intuitive/standard to compare to the uncompressed size (and have all ratios below 1.0). Also I don't understand the meaning of the comp./decomp. time in the left panel for uncompressed data.

(9) page 6, line 30: The paper could be much stronger with specific examples of individual variables and how affected by compression approach and choice of metric (e.g. by std. dev. or mean normalization). Since all results are averaged across datasets, this information is not available.

(10) Section 3: This section contains some useful information (and examples) about linear scaling and layer packing that would have been good to explain earlier in the paper when the concepts/algorithms are first introduced (and before the results are

given).

(11) More related lossy compression work on geophysical data should be mentioned for better context, for example:

Hubbe, Wegener et al., ISC '13 (http://link.springer.com/chapter/10.1007%2F978-3-642-38750-0_26)

Baker, et al., HPDC '14 (http://dl.acm.org/citation.cfm?id=2600217)

Woodring et al., LDAV '11 (http://ieeexplore.ieee.org/xpls/abs_all.jsp?arnumber=6092314&tag=1)

(12) Other competitive lossy compression algorithms for scientific data should probably be mentioned as many may be affected by differences in the variation across spatial dimensions for gridded data - this could be really interesting. Also many lossy compression methods for scientific data could eventually by incorporated into netcdf.

(12) Fig. 2: Because the differences between the datasets are not more thoroughly addressed, then it's unclear what conclusion to draw by comparing the SD and mean normalizations in Figure 2 (e.g., what is the takeaway point?). Basically, it seems that the two plots are quantitatively similar enough that both should be included only to illustrate a point, which I am not seeing. Can you clarify?

(13) fig 3: Same comment as above, plus I am not sure what conclusion to draw given that some datasets compress better than others without a more clear understanding of dataset differences. I think looking at individual variables, rather than entire datasets would make it easier for the reader to understand the differences in the approaches.

———————————- Final thoughts ———————————-

I like the idea of treating spatial dimensions differently with lossy compression, and I think the authors could have really taken off with this concept and it explored it much more thoroughly. I question whether the contributions in this particular version are significant enough for a GMD paper.

---

## Referee Comment (RC2) · Anonymous Referee #2 · 30 Aug 2016

This paper describes a variant of lossy encoding which leverages the multi-dimensional nature of many scientific datasets that have greater data variances along different axes. The axes with small variations in data values are labeled "thin dimensions" and the axes with large variations in values are labeled "thick dimensions". The datasets are then "layer packed" with a linear scaling algorithm in the thin dimensions, recording a scale & offset value for each coordinate in the thick dimension.

I think the insights into the "thick" and "thin" dimensions are the primary value of this paper, with the actual compression algorithm and results being less important, overall. Applying the idea of thick & thin dimensions appropriately to other compression methods (such as the JPEG-2000 algorithm used in GRiB2) would be more valuable than

just the idea of the simple scale & offset compression chosen.

Near the bottom of page 6, "for simplicity will have" should be corrected to "for simplicity we have".
* * *

---

## Referee Comment (RC3) · Anonymous Referee #2 · 30 Aug 2016

Very nice review, much more detailed than mine. We seem to have homed in on the same insights: the differences in dimensions are the valuable part of the paper, and they aren't explored in enough detail to warrant a lot of enthusiasm in the current state of the paper.

My current feeling is a very "weak" accept, and I would prefer to ask for further exploration of the dimension ideas.

**[Printer-friendly version](...)**

**[Discussion paper](...)**

---

## Referee Comment (RC4) · Anonymous Referee #3 · 1 Sep 2016

—Summary—

The paper introduces a "layer packing" lossy compression technique that takes advantage of the minimal horizontal variations in geoscience data relative to the larger variations across vertical dimensions. The layer packing technique is compared against many widely used lossless and lossy compression techniques and evaluated based on accuracy and time to solution. Layer packing is found to be beneficial in some cases while not in others, leading to the conclusion that care must be taken to evaluate whether lossy compression is worth the risk.

—General Comments—

The paper makes a good first attempt to evaluate the layer packing technique, but the paper would benefit from an additional revision. First, it's not clear what the paper is contributing. The authors state that the technique is used in GRIB (page 7, section 3) but that the evaluation was not possible due to relative error not being reported. Since the technique is not new, then the only contributions of the paper are the announcement of the general availability of the new non-GRIB tools, as well as the modestly detailed evaluation of the many compression techniques.

–Specific Comments and Technical Corrections—

The title, though catchy, is overloading the term "Goldilocks Zone" – the region around a star where perhaps liquid water might be found on a planet's surface. The title after the colon is clear on its own.

Page 2, line 3: "NetCDF" starts the last sentence on the line, though it should be "netCDF" for consistency.

Page 2, line 5: Why are three references necessary to describe the "deflate" compression method?

Throughout the paper, be consistent with terms. scale-offset vs scale and offset. linear-packing vs linear packing.

Page 3, line 30: I would suggest adding that ncdump is a command-line utility from the netCDF package because it might not be common knowledge. The paper introduces the "ncpacklayer" program and also uses other "nc"-prefixed tools from the NCO suite. For example, perhaps the following: "...(following the output format for the netCDF command-line utility ncdump)..."

Page 3 (section 2 in general): More detail could be spent on the layer packing technique itself; the many monospaced examples of section 2 don't substantially add to the narrative and instead come across like a tutorial or README.

Page 4, line 11: run-on sentence

Page 4: The dollar symbol "$" is not explained, though I think you meant for it to refer to a shell variable syntax.

Page 5, Section 2.3: If I do the math correctly, the size of the datasets are (1) 962MB, (2) 267MB, (3) 68MB, (4) 613MB, (5) 30MB, and (6) 717MB. The rationale for the proposed compression is the growing volume of data in the geosciences, though none of these datasets are over a gigabyte in size. Compression of a multi-gigabyte dataset would make the argument more compelling, because datasets of such size will become more commonplace. Writing large datasets to disk as they are computed is a challenging problem and it would be nice to evaluate whether compressing large datasets is a viable option as they are generated.

General comment about all Figures: Consider labeling the left and right panes of each figure as (a) and (b). For example, page 6, paragraphs starting on lines 9 and 17 sound too similar since Figure 1 is showing different things but is referred to in the text in the same way. It would be more clear to say something like "Figure 1A shows..." and "Figure 1B presents..."

Page 7: Starting on this page, for some reason all references to "figure 3" are lower case.

Page 8: Figure 1: The red and orange colors are too similar, though their position is clear from the legend.

Page 8, Figure 1, right panel: What does it mean to have the first column as "uncompressed" time since everything is normalized to DEFLATE? Was it the time to generate the data? Was it the time to copy the file?

Page 8, line 4: The reference to the HDF Group is used as an in-text citation as "(Group, 2016)". It would be best to fix your citation to not use HDF Group as a first/last name pair. See also your references on page 13, line 17.

Page 9, line 1: run-on sentence

Page 10, Figure 3 caption: capitalize the Figure 1 and Figure 2 references.

Page 11, line 6: misspelled "considered" – please consider a full spell check.
* * *

---

## Author Comment (AC1) · 28 Oct 2016

We wish to thank the reviewers to taking the time to read the manuscript and provide feedback. We note that we have taken the challenge of major revision seriously and reworked the analysis to a much more fine-grained level, included a range of new and interesting results, remade all the figures, and restructured and rewritten much of the text. We believe that the reviewers' comments have helped to improve the manuscript and strengthen our findings.

Please see the other replies which include the revised manuscript and a summary of changes.

[Figure]

**1   Reviewer 1**

**1.1   General comments**

1. *This paper addresses an important issue because data compression is very much needed to mitigate large data volumes in geophysical data. Treating the dimensions differently when applying lossy compression to gridded data makes a lot of sense.*

   We agree.

2. *Section 1 and 2 need some rearranging and improvement (more details are given below in "specific comments") in terms of introducing the ideas and terminology. It could be better to shorten the introduction and then really explain the methods well in section 2.*

   We have rearranged material in these sections given the feedback provided.

3. *The audience for this work may not be too familiar with compression techniques other than just using defaults in netCDF, so improving the explanations for the techniques would be helpful. (For example, defining a "deflate and shuffle" algorithm).*

   We have provided additional details as suggested.

4. *The paper's contribution should be clarified in the introduction (section 1). It is not clear to me whether "layer packing" is a new idea that is first presented here. (It is mentioned a bit more clearly in section 3).*

   Layer packing per say is not a new idea, and is the foundation for compression in the GRIB data format. However the idea of layer-packing is generalised here

beyond two-dimensional slices. The work presented here is a test-of-concept for combining some of the better aspects of both GRIB and netCDF/HDF5 formats. The introduction and discussion reiterate these points.

5. *For this paper to really impact the broader geophysical data community, I feel that more details on the compression approaches need to be provided.*

We have provided more details as recommended.

6. *More details on the datasets are needed to be able to understand why compression effects the each differently. Perhaps look at variables instead of multi-variable datasets?*

This is an excellent suggestion and one that we have adopted. One of the main changes to the manuscript between the initial submission and this revision is that we examine compression in a variable-by-variable approach rather than as a whole-dataset approach. This allows us to look at individual variables in terms of their compressibility, the "complexity" of the variable and error resulting from the lossy compression; this fine-grained approach allows for greater insight and a much larger sample size. As such the results section has been heavily revised.

**1.2 Specific comments**

1. *page 2, par. 1: For this audience, please give more explanation of the techniques. For example, please provide more explanation of how "deflate and shuffle" works (rather than just pointing to a reference).*

We have introduced additional detail about these methods as recommended.

2. *page 2, line 22: "Linear packing with a single scale-offset parameter" – is discussed here but not well-defined. Note that "packing" is later defined in line 32.*

*Then "scalar linear packing" on p.3. line 2. In general, the terminology used and defined in this paragraph is hard to follow in that it is sometimes defined after being used. (Also, is "linear packing with a single scale-offset parameter" the same as "scalar linear packing"?)*

We have reviewed how the notation is introduced in order to improve readability.

3. *p.2, line 29: I'm not sure the audience will be familiar with "quantization" (like the audience for a CS publication would).*

   This has been clarified

4. *section 2.1.1 ("Layer packing") Here I would suggest providing more detail (maybe an example) – particularly if this approach is the main contribution of the paper. Rather than providing syntax details, consider defining/explaining the parameters (the reader may not be familiar with what these are) here.*

   In hindsight we agree that details about the algorithm itself are required, rather than syntax. We have moved the syntax to a supplementary section. The algorithm itself is outlined in the methods section.

5. *section 2.1.2, line 15: Explain what "level" means in the algorithm.*

   This has been explained.

6. *section 2.1.2, line 17: Explain a shuffle filter.*

   We have added additional details.

7. *section 2.3: Regarding the datasets listed, more information about the model source (other than acronym and reference) would be helpful - especially in interpreting the results. Without more details, I cannot really understand how the*

*datasets differ and, therefore, why/how they would respond to compression differently. For example, the number of grid points are given - but does this number represent a domain on the entire globe for all datasets? The number of vertical levels is listed, but do all models simulate to the same height? What is the time dimension? Hourly? Monthly averages? Is the time dimension the same for each data set?*

The original description of these datasets was deliberately kept short, as this was not the main focus of the paper. We have compromised by abbreviating the description of the datasets to a table and moving the full descriptions of these datasets to the Supplementary Material section.
Regarding the question about why variables respond differently to compression, we believe that this has been solidly addressed in the analysis of the entropy of the data and exponent fields, which was made possibly by following the suggestion to shift the focus of the paper from compressing entire datasets to compressing individual variables.

8. *Fig 1: For compression results, I think it would be more intuitive/standard to compare to the uncompressed size (and have all ratios below 1.0). Also I don't understand the meaning of the comp./decomp. time in the left panel for uncompressed data.*

The compression ratios are now defined in terms of the uncompressed size as suggested, and we have also moved to a more standard definition of the compression ratio (i.e. uncompressed size / compressed size, so that larger values represent greater compression). The compression times represent the time taken from the original data to the compressed file, whereas the decompression time is to unpack the layer-packed data. This has been clarified

9. *page 6, line 30: The paper could be much stronger with specific examples of individual variables and how affected by compression approach and choice of*

[Figure]

*metric (e.g. by std. dev. or mean normalization). Since all results are averaged across datasets, this information is not available.*

We agree and we have adopted the variable-level rather than dataset-level approach. We included examples of six variables (among a total of 255) in the Supplementary Material document as illustrations of the errors induced by the six lossy compression methods considered.

10. *Section 3: This section contains some useful information (and examples) about linear scaling and layer packing that would have been good to explain earlier in the paper when the concepts/algorithms are first introduced (and before the results are given).*

We have given additional details about linear scaling and layer packing in the Methods section. Additional examples for illustrative variables appear in the Results section.

11. *More related lossy compression work on geophysical data should be mentioned for better context, for example: Hubbe, Wegener et al., ISC '13 (http:// link.springer.com/ chapter/ 10.1007%2F978-3-642-38750-0_26), Baker, et al., HPDC '14 (http:// dl.acm.org/ citation.cfm?id=2600217), Woodring et al., LDAV '11 (http:// ieeexplore.ieee.org/ xpls/ abs_all.jsp?arnumber=6092314&tag=1)*

We have given more details about related lossy compression work in this field. We thank the reviewer for the suggested citations and have included some in the manuscript.

12. *Other competitive lossy compression algorithms for scientific data should probably be mentioned as many may be affected by differences in the variation across spatial dimensions for gridded data – this could be really interesting. Also many lossy compression methods for scientific data could eventually by incorporated into netCDF.*

We have expanded the discussion to refer to other lossy compression algorithms for scientific data, formats beyond netCDF (e.g. based on image- and video-compression).

13. *Fig. 2: Because the differences between the datasets are not more thoroughly addressed, then it's unclear what conclusion to draw by comparing the SD and mean normalizations in Figure 2 (e.g., what is the takeaway point?). Basically, it seems that the two plots are quantitatively similar enough that both should be included only to illustrate a point, which I am not seeing. Can you clarify?*

Both plots were included in order avoid the perception of a biased interpretation of the results. Normalization by the SD or the mean advantages one method or the other, however the conclusions are the same regardless of the normalization. We agree that including both plots does not add much value to the paper. We note that all the figures have been completely reworked.

14. *fig 3: Same comment as above, plus I am not sure what conclusion to draw given that some datasets compress better than others without a more clear understanding of dataset differences. I think looking at individual variables, rather than entire datasets would make it easier for the reader to understand the differences in the approaches.*

As noted previously, we agree with the reviewer's comment and have redone the analysis to examine variables separately, rather than groups of variables clustered together as datasets.

**1.3 Final thoughts**

1. *I like the idea of treating spatial dimensions differently with lossy compression, and I think the authors could have really taken off with this concept and it explored*

*it much more thoroughly. I question whether the contributions in this particular version are significant enough for a GMD paper.*

The purpose of this study was to test the concept of layer-packing, in an attempt to combine some of the best aspects of the GRIB and netCDF/HDF5 data formats. We acknowledge that the results have not been conclusively in favour of the layer-packing with respect to bit-grooming, however we would argue that this is worth publishing all the same. This partly relates to the discussion of publishing "positive" versus "negative" results; if only "positive" findings are published, this will result in a great deal of time and effort being wasted within the scientific community in repeating superficially appealing experiments. As such, transferring this knowledge to the public domain has value. The geoscientific modelling and measurement community (e.g. the volume of data generated by satellite retrievals) relies heavily on these data formats, and it is important that their refinement is an ongoing process.

Regardless of any ambiguity between the choice of bit-grooming or layer-packing, one clear result from this study is that simple linear packing typically results in *much* greater loss of precision than either of the two lossy methods discussed here. This is despite its widespread use.

Other useful contributions include the focus on the error-compression trade-off, the finding that the normalized entropy of the exponent field can be used to help determine which compression method is most appropriate, and the idea (introduced in the discussion) that the changes in the normalized entropy of the data could be used to determine how many significant figures should be retained.

---

## Author Comment (AC2) · 28 Oct 2016

We wish to thank the reviewers to taking the time to read the manuscript and provide feedback. We note that we have taken the challenge of major revision seriously and reworked the analysis to a much more fine-grained level, included a range of new and interesting results, remade all the figures, and restructured and rewritten much of the text. We believe that the reviewers' comments have helped to improve the manuscript and strengthen our findings.

Please see the other replies which include the revised manuscript and a summary of changes.

[Figure]

**1 Reviewer 2**

**1.1 General comments**

1. *This paper describes a variant of lossy encoding which leverages the multi-dimensional nature of many scientific datasets that have greater data variances along different axes. The axes with small variations in data values are labeled "thin dimensions" and the axes with large variations in values are labeled "thick dimensions". The datasets are then "layer packed" with a linear scaling algorithm in the thin dimensions, recording a scale & offset value for each coordinate in the thick dimension.*
*I think the insights into the "thick" and "thin" dimensions are the primary value of this paper, with the actual compression algorithm and results being less important, overall.*

   Yes – one of the main things we are trying to do here is to assess whether treating different dimensions differently during gives much benefit over and above other methods that can be easily applied to such datasets. This is essentially trying to combine the best elements of GRIB and netCDF/HDF5.

**1.2 Specific comments**

1. *Applying the idea of thick & thin dimensions appropriately to other compression methods (such as the JPEG-2000 algorithm used in GRIB2) would be more valuable than just the idea of the simple scale & offset compression chosen.*

   We agree, and we spent a large amount of time trying to get this to work while preparing these revisions.
   In revising this work, we were able to run (after many technical hiccups) the same
set of tests for GRIB/JPEG2000 compression as well (using 8, 12, 16 and 20 bits to represent the data). Our preliminary results showed that the JPEG2000 algorithm yields greater compression compared to the methods presented here for the same level of error; this echoes the findings of Caron (2014, www.ecmwf.int/sites/default/files/elibrary/2014/13711-converting-grib-netcdf-4.pdf), which describe the efficient compression achieved. However like bit-grooming or layer-packing, JPEG2000 does not offer clear controls about the resultant errors and thus some experimentation (in setting the number of bits per value) is needed to avoid excessive loss of precision. We found that there was a large spread in the magnitude of the relative errors compared to the other methods considered.

However the technical challenges required to convert a general netCDF field into GRIB format to be far in excess of what may be recommended to the average practitioner of geoscientific modelling. For this reason, and for the large spread in the compression and error result in the GRIB-compressed fields, and in order to keep the manuscript as focussed as possible, we chose not to include these results.

2. *Near the bottom of page 6, "for simplicity will have" should be corrected to "for simplicity we have".*

Yes, well spotted. We have fixed this.

---

## Author Comment (AC3) · 28 Oct 2016

We wish to thank the reviewers to taking the time to read the manuscript and provide feedback. We note that we have taken the challenge of major revision seriously and reworked the analysis to a much more fine-grained level, included a range of new and interesting results, remade all the figures, and restructured and rewritten much of the text. We believe that the reviewers' comments have helped to improve the manuscript and strengthen our findings.

Please see the other replies which include the revised manuscript and a summary of changes.

[Figure]

**1 Reviewer 2's comments to Review 1**

1. *Very nice review, much more detailed than mine. We seem to have homed in on the same insights: the differences in dimensions are the valuable part of the paper, and they aren't explored in enough detail to warrant a lot of enthusiasm in the current state of the paper. My current feeling is a very "weak" accept, and I would prefer to ask for further exploration of the dimension ideas.*

   Following the suggestions from Reviewer 1, the analysis and results have been considerably expanded and the revised manuscript offers further perspectives into the relationship between lossy compression, the resulting error and underlying complexity of the data.

   ──────────────────────────────

---

## Author Comment (AC4) · 28 Oct 2016

We wish to thank the reviewers to taking the time to read the manuscript and provide feedback. We note that we have taken the challenge of major revision seriously and reworked the analysis to a much more fine-grained level, included a range of new and interesting results, remade all the figures, and restructured and rewritten much of the text. We believe that the reviewers' comments have helped to improve the manuscript and strengthen our findings.

Please see the other replies which include the revised manuscript and a summary of changes.

**1 Reviewer 3**

**1.1 Summary**

*The paper introduces a "layer packing" lossy compression technique that takes advantage of the minimal horizontal variations in geoscience data relative to the larger variations across vertical dimensions. The layer packing technique is compared against many widely used lossless and lossy compression techniques and evaluated based on accuracy and time to solution. Layer packing is found to be beneficial in some cases while not in others, leading to the conclusion that care must be taken to evaluate whether lossy compression is worth the risk.*

**1.2 General Comments**

1. *The paper makes a good first attempt to evaluate the layer packing technique, but the paper would benefit from an additional revision. First, it's not clear what the paper is contributing. The authors state that the technique is used in GRIB (page 7, section 3) but that the evaluation was not possible due to relative error not being reported. Since the technique is not new, then the only contributions of the paper are the announcement of the general availability of the new non-GRIB tools, as well as the modestly detailed evaluation of the many compression techniques.*

   The geoscientific modelling and remote-sensing community has to deal with the ever-growing volume of data generated. As such, it is important that the storage methods are reviewed in terms of the trade-off between compression, error and read/write times.
   We have tried to avoid debate about data formats. Both have an important roles; the geoscientific community relies heavily on netCDF/HDF5, and GRIB remains

the format of choice in many operational meteorological centres. Despite its excellent compression performance, GRIB can be regarded as less user-friendly.
The GRIB layer-packing is restricted to two-dimensional slices, whereas the layer-packing described here can operate on arbitrary hyperslices. The work presented in this manuscript aims to generate discussion about ways of incorporating the best of both methods.

With reference to the comment from Page 7, Section 3: "Caron (2014) estimated that GRIB2 files are on average 44% of the size of the equivalent deflate-compressed netCDF-4 files (n.b. relative errors were not reported, which limits the comparison)". The intended meaning was that the study of Caron (2014) reported the compression ratio, but not the relative errors, which makes it difficult to place the Caron (2014) results with those of this study.

The revisions to the original manuscript, focusing the analysis on the compressibility, errors and complexity of individual variables offers additional insights into these relationship and we believe adds substantially to the value of the paper.

**1.3 Specific Comments and Technical Corrections**

1. *The title, though catchy, is overloading the term "Goldilocks Zone" – the region around a star where perhaps liquid water might be found on a planet's surface. The title after the colon is clear on its own.*

   We have abbreviated the title, which as already been through several iterations.

2. *Page 2, line 3: "NetCDF" starts the last sentence on the line, though it should be "netCDF" for consistency.*

   We have revised for consistency of this term.

3. *Page 2, line 5: Why are three references necessary to describe the "deflate"*

*compression method? Throughout the paper, be consistent with terms. scale-offset vs scale and offset. linear-packing vs linear packing.*

Additional description of the deflate and shuffle algorithms has been added as suggested by Reviewer 1. We have reconsidered the references in this section. We have also reviewed the usage of the terms mentioned to improve the consistency of the manuscript.

4. *Page 3, line 30: I would suggest adding that ncdump is a command-line utility from the netCDF package because it might not be common knowledge. The paper introduces the "ncpacklayer" program and also uses other "nc"-prefixed tools from the NCO suite. For example, perhaps the following: "...(following the output format for the netCDF command-line utility ncdump)..."*

Yes, this is correct, thanks for pointing this out. We have clarified this point.

5. *Page 3 (section 2 in general): More detail could be spent on the layer packing technique itself; the many monospaced examples of section 2 don't substantially add to the narrative and instead come across like a tutorial or README.*

We have expanded the description of the algorithm itself. To keep the article short and concise, we have moved these details to an appendix.

6. *Page 4, line 11: run-on sentence*

Thanks for pointing this out. This has been corrected.

7. *Page 4: The dollar symbol "$" is not explained, though I think you meant for it to refer to a shell variable syntax.*

Yes, this is correct. This has been clarified.

8. *Page 5, Section 2.3: If I do the math correctly, the size of the datasets are (1) 962MB, (2) 267MB, (3) 68MB, (4) 613MB, (5) 30MB, and (6) 717MB. The rationale for the proposed compression is the growing volume of data in the geosciences, though none of these datasets are over a gigabyte in size. Compression of a multi-gigabyte dataset would make the argument more compelling, because datasets of such size will become more commonplace. Writing large datasets to disk as they are computed is a challenging problem and it would be nice to evaluate whether compressing large datasets is a viable option as they are generated. General comment about all Figures: Consider labeling the left and right panes of each figure as (a) and (b). For example, page 6, paragraphs starting on lines 9 and 17 sound too similar since Figure 1 is showing different things but is referred to in the text in the same way. It would be more clear to say something like "Figure 1A shows..." and "Figure 1B presents..."*

The point about the magnitude of the file size is quite reasonable. We ran the test suite on variable of size 1.5 GB to examine the performance of the methods on larger datasets. This was included as an example referenced in the timing results, rather than adding it to the suite of variables presented in all results. This was mainly because, in the process of setting it up, the test suite was run many dozens of times; to accelerate the testing the variables considered were kept relatively small (the largest was about 65 MB).
However by the same token, the analysis for the revised manuscript has been done on individual variables alone, so the basic unit of study has become much smaller. While this might not impress those working with terabyte-scale data, it allows for greater insight into the methodology itself.
Regarding the figures, some of these have been moved to a supplementary material section. All panel plots now have labels (a), (b), etc., as suggested.

9. *Page 7: Starting on this page, for some reason all references to "figure 3" are lower case.*

Thanks for pointing this out – it has been fixed.

10. *Page 8: Figure 1: The red and orange colors are too similar, though their position is clear from the legend.*

   All the figures have been thoroughly reworked. The color scheme in question no longer appears.

11. *Page 8, Figure 1, right panel: What does it mean to have the first column as "uncompressed" time since everything is normalized to DEFLATE? Was it the time to generate the data? Was it the time to copy the file?*

   Yes, in hindsight this wasn't very clear. It was effectively the time to copy the data. This bar is not included it in the revised manuscript. Thanks for drawing attention to it.

12. *Page 8, line 4: The reference to the HDF Group is used as an in-text citation as "(Group, 2016)". It would be best to fix your citation to not use HDF Group as a first/last name pair. See also your references on page 13, line 17.*

   Thanks for pointing this out. The default behaviour of the reference manager should have been over-ruled. This has been corrected.

13. *Page 9, line 1: run-on sentence*

   Thanks for pointing this out. It has been corrected.

14. *Page 10, Figure 3 caption: capitalize the Figure 1 and Figure 2 references.*

   This has been made more consistent.

15. *Page 11, line 6: misspelled "considered" – please consider a full spell check.*

This has been fixed and we will ensure to run the spell checker again before resubmitting.

---

## Author Comment (AC5) · 28 Oct 2016

We wish to thank the reviewers to taking the time to read the manuscript and provide feedback. We note that we have taken the challenge of major revision seriously and reworked the analysis to a much more fine-grained level, included a range of new and interesting results, remade all the figures, and restructured and rewritten much of the text. We believe that the reviewers' comments have helped to improve the manuscript and strengthen our findings.

Please find attached a PDF document that collates the revised draft, the new supplementary material document and the full point-by-point reply to the reviewers.

**Main changes**

- Compression, errors and complexity are assessed at the variable-level, rather than the dataset-level (i.e. for a number of variables combined).

- We calculated a range of statistics on the individual variables, in order to improve our understanding of why certain variables compress well with one method or another.

- Some material was moved to a supplementary document.

- The introduction has been abbreviated as recommended.

- The Methods section was expanded to provide a clearer description of the layer-packing method.

- The discussion includes a brief review of related work.

- Additional description of the deflate and shuffle compression algorithms were added to the Methods section.

- All figures have been reworked.

Minor changes

- Variables are now chunked in a consistent manner for the different methods to improve comparability across compression methods.

- A minor error was found and corrected in the calculation of file sizes. The differences would have been very minor for the results in the original manuscript, since the file sizes were much larger than when doing the analysis on individual variables, but became apparent when working with the single-variable data files.

**[GMDD](https://www.geosci-model-dev-discuss.net/)**

Interactive
comment

The error was that the results were calculated based on "resident" rather than "actual" file size.

- Minor improvements were made to the layer-packing code, resulting in more stable treatment of non-finite values, avoiding rare cases of floating-point overflow, and more stable handling of dimensions.

- We ran the test suite on a variable of size 1.5 GB to examine the performance of the methods on larger datasets. This was included as an example referenced in the timing results, rather than adding it to the suite of variables presented in all results. This was mainly because, in the process of setting it up, the test suite was run many dozens of times; to accelerate the testing the variables considered were kept relatively small (the largest was about 65 MB).

Please also note the supplement to this comment:
http://www.geosci-model-dev-discuss.net/gmd-2016-177/gmd-2016-177-AC5-supplement.pdf

**Supplement:**

[revised manuscript text omitted]

October 28, 2016

**1  Compression methods**

**1.1  Command-line calls used**

The compression methods compared were realized using the commands listed below. For each method apart from LAY, command-line tools from the NCO bundle (Zender, 2008) were used.

1. DEFLATE: Deflate compression (level 4) with shuffle filter

   ```
   ncks -4 -L4 in.nc out.nc
   ```

2. NSD2, NSD3, NSD4, NSD5: Deflate compression (level 4) with shuffle filter, and bit grooming storing 2, 3, 4, or 5 significant figures (respectively). The following yields three significant digits (NSD3).

   ```
   ncks -4 -L4 --ppc $var=3 in.nc out.nc
   ```

3. LIN: Deflate compression (level 4) with shuffle filter, scalar linear packing for each variable

   ```
   ncpdq -4 -L4 in.nc out.nc
   ```

4. LAY: Deflate compression (level 4) with shuffle filter, layer packing for selected dimensions

   ```
   ncpacklayer -L4 -v $var -d $dims in.nc out.nc
   ```

In the above, `$var` is a Linux/Unix shell variable giving the name of the sole variable contained within the input file.

We note that the above omits details of how the handling of chunking of variables was controlled. This was done by repeating, for each thick dimension, the argument `--cnk_dmn $dim,1` with the shell variable `$dim` set to the name of the dimension.

**1.2  Further details about ncpacklayer**

The compression is performed as follows:

```
ncpacklayer -d thickdim1,thickdim2 -v var1,var2,var3 original.nc packed.nc
```

Other optional flags allow for increased verbosity (`-V`), over-writing existing output files (`-O`) and defining the DEFLATE compression level (`-L`).

The mandatory `-d` flag is followed by a comma-separated list of the thick dimensions. The optional `-v` flag is followed by a comma-separated list of variables to pack. The default is to pack all variables defined along any of the thick dimensions listed. In the output file (in this example `packed.nc`) each variable that is packed (e.g. `var1`) is replaced by a trio of variables containing the arrays of packed values, scale factors and offsets. In this example, these are termed `var1__short`, `var1__scale` and `var1__offset`, with data type unsigned short (i.e. two-byte) integer, floating-point and floating-point, respectively. Suppose the original definition of `var1` is (following output format for the command line utility `ncdump`, which is provided when the netCDF API rather than the NCO bundle):

```
float var1(thindim1, thickdim1, thickdim2, thindim2) ;
```

then the corresponding trio will have dimensions as follows:

```
ushort var1__short(thindim1, thickdim1, thickdim2, thindim2) ;
float var1__scale(thickdim1, thickdim2) ;
float var1__offset(thickdim1, thickdim2) ;
```

In other words, the scale and offset arrays have one element per thin slice. Data remain in netCDF format in this packed format and retain all their attributes. Data can be unpacked as follows:

```
ncunpacklayer packed.nc unpacked.nc
```

The `-d` and `-v` flags are not used, since this information is contained in the trios of packed arrays.

**2  Datasets**

The tests described above were applied to the following datasets. In each case, we have provided the full list of variables in the analysis since in some cases not all variables provided in the files were featured in the analysis. We have not gone so far as to describe each of variables listed below, since this would take up much more space and because this information can generally be found within the metadata of each data set (links are provided in each case).

1. ERA-Interim reanalysis data (Dee et al., 2011)

   - Filename: `ei_mnth_an_pl_15x15_90N0E90S3585E_20080901_20081201`
   - Horizontal domain: a regular latitude-longitude grid covering the globe at 1.5° resolution. Latitude dimension of length 121, longitude dimension of length 240.
   - Vertical dimension: 37 pressure levels ranging from 1000 hPa to 1 hPa
   - Time dimension: 16 six-hourly snap-shots ranging from 2008-01-09 00:00 UTC to 2008-01-12 18:00 UTC
   - Notes: Converted from GRIB format prior to the analysis.
   - Layer packing: Thick dimensions chosen to be the time and vertical level.
   - What do the variables describe: atmospheric dynamics, temperature, ozone mixing ratio, cloud properties, humidity
   - Variables: 14 variables were included in the analysis. These were: PV_GDS0_ISBL_S123, Z_GDS0_ISBL_S123, T_GDS0_ISBL_S123, U_GDS0_ISBL_S123, V_GDS0_ISBL_S123, Q_GDS0_ISBL_S123, W_GDS0_ISBL_S123, VO_GDS0_ISBL_S123, D_GDS0_ISBL_S123, R_GDS0_ISBL_S123, O3_GDS0_ISBL_S123, CLWC_GDS0_ISBL_S123 CIWC_GDS0_ISBL_S123, CC_GDS0_ISBL_S123
   - Availability: This dataset could not be distributed with the other files due to licensing restrictions but can be accessed through the ECMWF's public dataset portal (`http://apps.ecmwf.int/datasets/`), using the following set of inputs: stream = synoptic monthly means, vertical levels = pressure levels (all 37 layers), parameters = all 14 variables, dataset = interim_mnth, step = 0, version = 1, time = 00:00:00, 06:00:00, 12:00:00, 18:00:00, date = 20080901 to 20081201, grid = 1.5° × 1.5°, type = analysis, class = ERA Interim.

   A limited area subset from global MOZART model output (Brasseur et al., 1998). Dimensions: 9 × 10 grid-points in the horizontal, 56 vertical levels, 172 time-points. 77 variables with these four dimensions.

   - Filename: `mozart4geos5_2011-02-01_2011-03-16.nc`
   - Horizontal domain: a limited area subset of a global domain covering Australia. The global domain appears to have 95 × 144 gridpoints, while only 9 × 10 grid-points (lon × lat) in the horizontal are covered in this file. The grid spacing is regular at 2.5° resolution in the latitude dimension and 1.895° resolution in the longitude dimension.
   - Vertical dimension: fixed pressure levels ranging with mid-points ranging from 992.5 Pa to 1.868 Pa.
   - Time dimension: 172 temporal snapshots at six-hourly resolution ranging from 2011-02-01 06:00 UTC to 2011-02-01 12:00 UTC.
   - Notes: Originally downloaded through through the web-page `http://www.acom.ucar.edu/wrf-chem/mozart.shtml`. This file contained smaller variables (other than coordinate variables) that were not included in the analysis due to their relatively small size.
   - Layer packing: Thick dimension chosen to be the vertical level.
   - What do the variables describe: volume mixing ratios of many trace gases, mass mixing ratios of some aerosol classes, atmospheric dynamics, temperature, photolytic reaction rates

- Variables: 77 variables were included in the analysis. Their names were: BIGALD_VMR_inst, BIGALK_VMR_inst, BIGENE_VMR_inst, C10H16_VMR_inst, C2H2_VMR_inst, C2H4_VMR_inst, C2H5OH_VMR_inst, C2H6_VMR_inst, C3H6_VMR_inst, C3H8_VMR_inst, CB1_VMR_inst, CB2_VMR_inst, CH2O_VMR_inst, CH3CHO_VMR_inst, CH3CN_VMR_inst, CH3COCH3_VMR_inst, CH3COCHO_VMR_inst, CH3COOH_VMR_inst, CH3COOOH_VMR_inst, CH3O2_VMR_inst, CH3OH_VMR_inst, CH3OOH_VMR_inst, CH4_VMR_inst, CO_VMR_inst, CRESOL_VMR_inst, DMS_VMR_inst, DUST1, DUST2, DUST3, DUST4, GLYALD_VMR_inst, H2O, H2O2_VMR_inst, HCN_VMR_inst, HCOOH_VMR_inst, HNO3_VMR_inst, HO2NO2_VMR_inst, HO2_VMR_inst, HYAC_VMR_inst, HYDRALD_VMR_inst, ISOPNO3_VMR_inst, ISOP_VMR_inst, MACR_VMR_inst, MEK_VMR_inst, MPAN_VMR_inst, MVK_VMR_inst, N2O5_VMR_inst, N2O_VMR_inst, NH3_VMR_inst, NH4NO3_VMR_inst, NH4_VMR_inst, NO2_VMR_inst, NO3_VMR_inst, NOX, NOY, NO_VMR_inst, O3_VMR_inst, OC1_VMR_inst, OC2_VMR_inst, OH_VMR_inst, ONITR_VMR_inst, ONIT_VMR_inst, PAN_VMR_inst, Q, SA1_VMR_inst, SA2_VMR_inst, SA3_VMR_inst, SA4_VMR_inst, SO2_VMR_inst, SO4_VMR_inst, SOA_VMR_inst, T, TOLUENE_VMR_inst, U, V, jno2_rcon_inst, jo1d_rcon_inst

- Availability: available online at `https://figshare.com/projects/Layer_Packing_Tests/14480`

2. Model output from the Weather Research and Forecasting (WRF) model (Skamarock et al., 2005).

   - Filename: `wrfout_d03_2013-01-24_07:00:00`

   - Horizontal domain: A limited area domain over the city of Sydney and surrounding areas (Australia), including a portion over the sea. A Lambert Conformal map projection was used and the horizontal resolution was 1 km in the east-west and north-south dimensions. There were 165 grid-points in the east-west dimension and 140 in the north-south dimension.

   - Vertical dimension: 32 levels using a terrain-following, hydrostatic pressure coordinate from the surface to 5 hPa.

   - Time dimension: A single time snaps-hot (2013-01-24 07:00 UTC)

   - Notes: Model output from simulations by J. Silver. This file contained smaller variables (other than coordinate variables) that were not included in the analysis due to their relatively small size.

   - Layer packing: Thick dimension chosen to be the vertical level.

   - What do the variables describe: atmospheric dynamics, temperature, cloud properties, humidity

   - Variables: 20 variables were included in the analysis. Their names were: U, V, W, PH, PHB, T, P, PB, QVAPOR, QCLOUD, QRAIN, QICE, QSNOW, QNICE, QNSNOW, QNRAIN, QNDROP, TKE_PBL, EL_PBL, CLDFRA

   - Availability: available online at `https://figshare.com/projects/Layer_Packing_Tests/14480`

3. MERRA reanalysis product (Rienecker et al., 2011).

   - Filename: `MERRA300.prod.assim.inst3_3d_asm_Cp.20130601.nc`

   - Horizontal domain: a regular latitude-longitude grid covering the globe at 1.25° resolution. Latitude dimension of length 144, longitude dimension of length 288.

   - Vertical dimension: 37 pressure levels ranging from 1000 hPa to 0.1 hPa

   - Time dimension: 8 temporal snapshots at three-hourly frequency, ranging from 2013-06-01 00:00 UTC to 2013-06-01 21:00 UTC

   - Notes: This file contained smaller variables (other than coordinate variables) that were not included in the analysis due to their relatively small size.

   - Layer packing: Thick dimension chosen to be the vertical level.

   - What do the variables describe: atmospheric dynamics, temperature, cloud properties, humidity, ozone mixing ratio

   - Variables: 11 variables were included in the analysis. Their names were: EPV, H, O3, OMEGA, QI, QL, QV, RH, T, U, V

   - Availability: available online at `https://figshare.com/projects/Layer_Packing_Tests/14480`

4. Output of the mineral Dust Entrainment And Deposition (DEAD) model (Zender et al., 2003).

   - Filename: `dstmch90_clm.nc`

   - Horizontal domain: a regular latitude-longitude grid covering the globe at 1.875° resolution in the longitude dimension and 1.904° resolution in the latitude dimension. Latitude dimension of length 94, longitude dimension of length 192.

- Vertical dimension: a hybrid vertical coordinate system with 28 levels ranging from 1000 hPa to 2.7 hPa.

- Time dimension: one time-point

- Notes: This file contained smaller variables (other than coordinate variables) that were not included in the analysis due to their relatively small size.

- Layer packing: Thick dimension chosen to be the vertical level.

- What do the variables describe: atmospheric dynamics, temperature, cloud properties, humidity, mass and mass flux rates for dust (either total or in size different bins)

- Variables: 15 variables were included in the analysis. Their names were: U, V, T, Q, RELHUM, CLOUD, CWAT, DSTQ, DSTQ01, DSTQ02, DSTQ03, DSTQ04, DSTSSPCP, DSTSSEVP, DSTSS-DRY

- Availability: available at the DEAD model homepage (`http://dust.ess.uci.edu/dead/`) and also at `https://figshare.com/projects/Layer_Packing_Tests/14480`

5. Model output from the coupled numerical weather prediction and chemistry transport model CAM-SE (Dennis et al., 2012).

  - Filename: `famipc5_ne30_v0.3_00003.cam.h0.1979-01-L5.nc`

  - Horizontal domain: A non-rectangular cube-sphere mesh, ordered as a single array of 48602

  - Vertical dimension: a hybrid vertical coordinate system with 30 levels ranging from 992 hPa to 3.6 hPa.

  - Time dimension: Only a single time-point is represented

  - Notes: This file contained smaller variables (other than coordinate variables) that were not included in the analysis due to their relatively small size.

  - What do the variables describe: aerosol and trace-gas concentrations, atmospheric dynamics, temperature, cloud properties

  - Variables: 118 variables were included in the analysis. Their names were: AQRAIN, AQSNOW, AREI, AREL, AWNC, AWNI, CCN3, CLDICE, CLDLIQ, CLOUD, DCQ, DMS, DTCOND, DTV, FICE, FREQI, FREQL, FREQR, FREQS, H2O2, H2SO4, ICIMR, ICWMR, IWC, LIQCLDF, NU-MICE, NUMLIQ, OMEGA, OMEGAT, Q, QRL, QRS, RELHUM, SO2, SO2_XFRC, SOAG, T, U, UU, V, VD01, VQ, VT, VU, VV, Vbc_a1, Vdst_a1, Vdst_a3, V ncl_a1, Vncl_a2, Vncl_a3, Vpom_a1, Vso4_a1, Vso4_a2, Vso4_a3, Vsoa_a1, Vsoa_a2, WSUB, XPH_LWC, Z3, bc_a1, bc_a1_2, bc_a1_XFRC, bc_c1, dgnd_a01, dgnd_a02, dgnd_a03, dgnumwet1, dgnumwet2, dgnumwet3, dgnw_a01, dgnw_a02, dgnw_a03, dst_a1, dst_a1_2, dst_a3, dst_a3_2, dst_c1, dst_c3, ncl_a1, ncl_a1_2, ncl _a2, ncl_a2_2, ncl_a3, ncl_a3_2, ncl_c1, ncl_c2, ncl_c3, num_a1, num_a2, num_a3, num_c1, num_c2, num_c3, pom_a1, pom_a1_2, pom_a1_XFRC, pom_c1, so4_a1, so4_a1_2, so4_a1_XFRC, so4_a2, so4_a2_2, so4_a2_XFRC, so4_a3, so4_a3_2, so4_c1, so4_c2, so4_c3, soa_a1, soa_a1_2, soa_a2, soa_a2_2, soa_c1, soa_c2, w at_a1, wat_a2, wat_a3

  - Availability: available online at `https://figshare.com/projects/Layer_Packing_Tests/14480`

As described in the manuscript (under the heading "Complexity statistics"), the variables were classified as "sparse" or "dense". Sparse variables were highly compressible, which was often due their non-trivial components being limited to a fraction of the data array. Sparse variables were chosen to be those satisfying any one of the following conditions: the compression ratio is greater than 5.0 using DEFLATE, the fraction of values equal to the most common value in the entire variable is greater than 0.2, and the fraction of hyperslices where all values were identical is great than 0.2. The breakdown among the different categories is given in Table 1.

| CompRatio > 5 | globalMaxP > 0.2 | propUniform > 0.2 | # vars |
|:---:|:---:|:---:|:---:|
| T | T | T | 19 |
| T | T | F | 0 |
| T | F | T | 0 |
| T | F | F | 0 |
| F | T | T | 16 |
| F | T | F | 39 |
| F | F | T | 0 |
| F | F | F | 181 |

Table 1: Number of variables fitting different "sparsity" criteria. Abbreviations: CompRatio = compression ratio using DEFLATE (level 4), globalMaxP = the fraction of values equal to the most common value in the entire variable, propUniform = the fraction of hyperslices where all values were identical, # vars = number of variables.

**3 Normalization errors**

Figure 1C of the main manuscript shows the distribution of normalized errors for the six lossy compression methods. In the main manuscript (under the heading "Error and compression metrics"), four different methods are described for normalizing the root mean-squared error. These were:

A. calculating the RMSE and standard deviation (SD) separately for each thin slice, and averaging the ratio RMSE/SD across thin slices;

B. taking the average across the per-slice RMSE and SD values, and then taking the ratio of these averages – that is, mean(RMSE)/mean(SD);

C. the same as A, except normalizing by the per-slice mean rather than the per-slice SD;

D. the same as B, except normalizing by the average of the per-slice means.

Figure 1C of the main manuscript shows the distribution of errors for normalization method A, and this is repeated in panel A of Figure 1 (this document); similarly, the distribution of methods B, C and D appear in their respectively-named panels of the same figure.

[Figure]

Figure 1: Distribution of errors with different normalization methods, plotted separately by dense variables and sparse variables (white and grey boxes, respectively). Top-left: normalized by the per-layer standard deviation. Top-right: normalized by the average of the per-layer standard deviations. Bottom-left: normalized by the per-layer mean. Bottom-right: normalized by the whole-variable mean.

**4    Entropy and compression for reduced-precision fields**

For the reduced-precision fields, we assessed relationship between the normalized entropy of the data field (NEDF) and the compression ratios. Figure 2 shows the compression ratios relative to the *uncompressed file sizes* whereas Figure 3 displays the compression ratios relative to the *DEFLATE-compressed file sizes*. Figure 2 presents the NEDF for each variable whereas Figure 3 plots the reduction in the NEDF due to the lossy filters.

[Figure]

Figure 2: Compression ratios for each of the lossy compression methods compared to the respective normalized entropy of each variable's data field; this accounts for quantization of the data field.

[Figure]

Figure 3: Compression ratios *relative to DEFLATE* for each of the lossy compression methods compared to the reduction in the normalized entropy due to the lossy compression.

**5 Vertical profiles of errors for selected variables**

Figures 4 and 5 illustrate, for six selected variables among the 255 considered, vertical profiles of the RMSE for each of the lossy compression methods. Figure 4 shows the *absolute* RMSE whereas Figure 5 displays the RMSE normalized by the corresponding per-level standard deviation. The six variables presented are:

1. U_GDS0_ISBL_S123: East-west wind velocity (units = m s$^{-1}$)

2. T: temperature (units = K)

3. P: perturbation pressure (units = Pa)

4. O3: Ozone mixing ratio (units = Kg·Kg$^{-1}$)

5. DSTQ: Total dust tendency due to settling and turbulence (units = Kg·Kg$^{-1}$·s$^{-1}$, positive when a sink to the gridcell))

6. dgnumwet1: Aerosol mode wet diameter (units = m)

[Figure]

Figure 4: Errors from the six lossy compression methods are shown as a function of vertical level for six variables (one from each dataset included). Also shown are the corresponding mean (of the absolute values) and standard deviation for the given variable. The errors were not normalized. Note that two scales are shown on the horizontal axis (at the bottom of each panel), the upper of which pertains to the errors and the lower scale to the mean and standard deviation. Also note the logarithmic scale on the $x$-axis.

[Figure]

Figure 5: Relative errors (normalizing by the per-layer standard deviation) from the six lossy compression methods are shown as a function of vertical level for six variables (the same variables as shown in Figure 4). Note the logarithmic scale on the $x$-axis.

**6 Details of the complexity statistics calculated**

As described in the manuscript, a range of statistics were calculated for every variable in the analysis. The following presents details of each of these. The statistics calculated were:

1. the normalized entropy of the floating point array,

2. the normalized entropy of the exponent array,

3. the normalized entropy of the mantissa array,

4. the fraction of values equal to the mode (i.e. the most common value in the hyperslice),

5. statistics representing the decay rate of singular values,

   This was calculated by calculating the singular-value decomposition of the two-dimensional slice, then finding the points at which the cumulative sum exceeded 0.5, 0.75, 0.9 or 0.95 times the total sum of the singular values; this was then represented as the fraction of the total number of singular values at which these points were reached (i.e. this yielded four separate statistics).

   Similar to the above, except searching for the fraction of the singular value beyond which the singular values fall below or 0.05, 0.1, 0.25, or 0.5 times the largest singular value (i.e. this also yields four statistics).

6. the spatial autocorrelation at fixed separation distances,

   This was calculated by estimating, for each separation distance up to 0.66 of the smaller array dimension in the two-dimensional slice, the correlation between a random sample of points separated by this distance (calculating distances by the Cartesian distance metric in terms of grid-spacing, rather than physical space). This then formed a scatter-plot of correlation versus distance, through which a locally-weighted scatter-plot smoother (LOWESS) curve was fitted (Cleveland, 1981). The points at which this curve fell below 0.95, 0.9, 0.75 or 0.5 were noted and these were represented as the fraction along the length of the smaller axis (i.e. this yielded four statistics).

   The above was done for separations in only the rows or columns, in which case the points at which the curve fell below the threshold were represented as the fraction along the length of the corresponding axis (i.e. this yielded eight statistics in total).

7. the mean (or mean of absolute values) divided by the standard deviation,

8. same as above, except for non-zero values only,

9. the range of the exponent field,

10. the standard deviation of the exponent field, and

11. the logarithm of the largest non-zero value divided by the smallest non-zero value.

As well as these, two global statistics were calculated:

1. the fraction of values equal to the mode (i.e. the most common value) in the entire variable and

2. the fraction of hyperslices where all values were identical.

**Reply to reviewers: "Finding the Goldilocks zone: Compression-error trade-off for large gridded datasets"**

Jeremy Silver, Charlie Zender

October 28, 2016

We wish to thank the reviewers to taking the time to read the manuscript and provide feedback. We note that we have taken the challenge of major revision seriously and reworked the analysis to a much more fine-grained level, included a range of new and interesting results, remade all the figures, and restructured and rewritten much of the text. We believe that the reviewers' comments have helped to improve the manuscript and strengthen our findings.

**Main changes**

- Compression, errors and complexity are assessed at the variable-level, rather than the dataset-level (i.e. for a number of variables combined).

- We calculated a range of statistics on the individual variables, in order to improve our understanding of why certain variables compress well with one method or another.

- Some material was moved to a supplementary document.

- The introduction has been abbreviated as recommended.

- The Methods section was expanded to provide a clearer description of the layer-packing method.

- The discussion includes a brief review of related work.

- Additional description of the deflate and shuffle compression algorithms were added to the Methods section.

- All figures have been reworked.

**Minor changes**

- Variables are now chunked in a consistent manner for the different methods to improve comparability across compression methods.

- A minor error was found and corrected in the calculation of file sizes. The differences would have been very minor for the results in the original manuscript, since the file sizes were much larger than when doing the analysis on individual variables, but became apparent when working with the single-variable data files. The error was that the results were calculated based on "resident" rather than "actual" file size.

- Minor improvements were made to the layer-packing code, resulting in more stable treatment of non-finite values, avoiding rare cases of floating-point overflow, and more stable handling of dimensions.

- We ran the test suite on a variable of size 1.5 GB to examine the performance of the methods on larger datasets. This was included as an example referenced in the timing results, rather than adding it to the suite of variables presented in all results. This was mainly because, in the process of setting it up, the test suite was run many dozens of times; to accelerate the testing the variables considered were kept relatively small (the largest was about 65 MB).

**1 Reviewer 1**

**1.1 General comments**

1. *This paper addresses an important issue because data compression is very much needed to mitigate large data volumes in geophysical data. Treating the dimensions differently when applying lossy compression to gridded data makes a lot of sense.*

   We agree.

2. *Section 1 and 2 need some rearranging and improvement (more details are given below in "specific comments") in terms of introducing the ideas and terminology. It could be better to shorten the introduction and then really explain the methods well in section 2.*

   We have rearranged material in these sections given the feedback provided.

3. *The audience for this work may not be too familiar with compression techniques other than just using defaults in netCDF, so improving the explanations for the techniques would be helpful. (For example, defining a "deflate and shuffle" algorithm).*

   We have provided additional details as suggested.

4. *The paper's contribution should be clarified in the introduction (section 1). It is not clear to me whether "layer packing" is a new idea that is first presented here. (It is mentioned a bit more clearly in section 3).*

   Layer packing per say is not a new idea, and is the foundation for compression in the GRIB data format. However the idea of layer-packing is generalised here beyond two-dimensional slices. The work presented here is a test-of-concept for combining some of the better aspects of both GRIB and netCDF/HDF5 formats.
   The introduction and discussion reiterate these points.

5. *For this paper to really impact the broader geophysical data community, I feel that more details on the compression approaches need to be provided.*

   We have provided more details as recommended.

6. *More details on the datasets are needed to be able to understand why compression effects the each differently. Perhaps look at variables instead of multi-variable datasets?*

   This is an excellent suggestion and one that we have adopted. One of the main changes to the manuscript between the initial submission and this revision is that we examine compression in a variable-by-variable approach rather than as a whole-dataset approach.

This allows us to look at individual variables in terms of their compressibility, the "complexity" of the variable and error resulting from the lossy compression; this fine-grained approach allows for greater insight and a much larger sample size. As such the results section has been heavily revised.

**1.2   Specific comments**

1. *page 2, par. 1: For this audience, please give more explanation of the techniques. For example, please provide more explanation of how "deflate and shuffle" works (rather than just pointing to a reference).*

   We have introduced additional detail about these methods as recommended.

2. *page 2, line 22: "Linear packing with a single scale-offset parameter" – is discussed here but not well-defined. Note that "packing" is later defined in line 32. Then "scalar linear packing" on p.3. line 2. In general, the terminology used and defined in this paragraph is hard to follow in that it is sometimes defined after being used. (Also, is "linear packing with a single scale-offset parameter" the same as "scalar linear packing"?)*

   We have reviewed how the notation is introduced in order to improve readability.

3. *p.2, line 29: I'm not sure the audience will be familiar with "quantization" (like the audience for a CS publication would).*

   This has been clarified

4. *section 2.1.1 ("Layer packing") Here I would suggest providing more detail (maybe an example) – particularly if this approach is the main contribution of the paper. Rather than providing syntax details, consider defining/explaining the parameters (the reader may not be familiar with what these are) here.*

   In hindsight we agree that details about the algorithm itself are required, rather than syntax. We have moved the syntax to a supplementary section. The algorithm itself is outlined in the methods section.

5. *section 2.1.2, line 15: Explain what "level" means in the algorithm.*

   This has been explained.

6. *section 2.1.2, line 17: Explain a shuffle filter.*

   We have added additional details.

7. *section 2.3: Regarding the datasets listed, more information about the model source (other than acronym and reference) would be helpful - especially in interpreting the results. Without more details, I cannot really understand how the datasets differ and, therefore, why/how they would respond to compression differently. For example, the number of grid points are given - but does this number represent a domain on the entire globe for all datasets? The number of vertical levels is listed, but do all models simulate to the same height? What is the time dimension? Hourly? Monthly averages? Is the time dimension the same for each data set?*

The original description of these datasets was deliberately kept short, as this was not the main focus of the paper. We have compromised by abbreviating the description of the datasets to a table and moving the full descriptions of these datasets to the Supplementary Material section.

Regarding the question about why variables respond differently to compression, we believe that this has been solidly addressed in the analysis of the entropy of the data and exponent fields, which was made possibly by following the suggestion to shift the focus of the paper from compressing entire datasets to compressing individual variables.

8. *Fig 1: For compression results, I think it would be more intuitive/standard to compare to the uncompressed size (and have all ratios below 1.0). Also I don't understand the meaning of the comp./decomp. time in the left panel for uncompressed data.*

   The compression ratios are now defined in terms of the uncompressed size as suggested, and we have also moved to a more standard definition of the compression ratio (i.e. uncompressed size / compressed size, so that larger values represent greater compression). The compression times represent the time taken from the original data to the compressed file, whereas the decompression time is to unpack the layer-packed data. This has been clarified

9. *page 6, line 30: The paper could be much stronger with specific examples of individual variables and how affected by compression approach and choice of metric (e.g. by std. dev. or mean normalization). Since all results are averaged across datasets, this information is not available.*

   We agree and we have adopted the variable-level rather than dataset-level approach. We included examples of six variables (among a total of 255) in the Supplementary Material document as illustrations of the errors induced by the six lossy compression methods considered.

10. *Section 3: This section contains some useful information (and examples) about linear scaling and layer packing that would have been good to explain earlier in the paper when the concepts/algorithms are first introduced (and before the results are given).*

    We have given additional details about linear scaling and layer packing in the Methods section. Additional examples for illustrative variables appear in the Results section.

11. *More related lossy compression work on geophysical data should be mentioned for better context, for example: Hubbe, Wegener et al., ISC '13 (`http://link.springer.com/chapter/10.1007%2F978-3-642-38750-0_26`), Baker, et al., HPDC '14 (`http://dl.acm.org/citation.cfm?id=2600217`), Woodring et al., LDAV '11 (`http://ieeexplore.ieee.org/xpls/abs_all.jsp?arnumber=6092314&tag=1`)*

    We have given more details about related lossy compression work in this field. We thank the reviewer for the suggested citations and have included some in the manuscript.

12. *Other competitive lossy compression algorithms for scientific data should probably be mentioned as many may be affected by differences in the variation across spatial dimensions for gridded data – this could be really interesting. Also many lossy compression methods for scientific data could eventually by incorporated into netCDF.*

    We have expanded the discussion to refer to other lossy compression algorithms for scientific data, formats beyond netCDF (e.g. based on image- and video-compression).

13. *Fig. 2: Because the differences between the datasets are not more thoroughly addressed, then it's unclear what conclusion to draw by comparing the SD and mean normalizations in Figure 2 (e.g., what is the takeaway point?). Basically, it seems that the two plots are quantitatively similar enough that both should be included only to illustrate a point, which I am not seeing. Can you clarify?*

Both plots were included in order avoid the perception of a biased interpretation of the results. Normalization by the SD or the mean advantages one method or the other, however the conclusions are the same regardless of the normalization. We agree that including both plots does not add much value to the paper. We note that all the figures have been completely reworked.

14. *fig 3: Same comment as above, plus I am not sure what conclusion to draw given that some datasets compress better than others without a more clear understanding of dataset differences. I think looking at individual variables, rather than entire datasets would make it easier for the reader to understand the differences in the approaches.*

As noted previously, we agree with the reviewer's comment and have redone the analysis to examine variables separately, rather than groups of variables clustered together as datasets.

**1.3 Final thoughts**

1. *I like the idea of treating spatial dimensions differently with lossy compression, and I think the authors could have really taken off with this concept and it explored it much more thoroughly. I question whether the contributions in this particular version are significant enough for a GMD paper.*

The purpose of this study was to test the concept of layer-packing, in an attempt to combine some of the best aspects of the GRIB and netCDF/HDF5 data formats. We acknowledge that the results have not been conclusively in favour of the layer-packing with respect to bit-grooming, however we would argue that this is worth publishing all the same. This partly relates to the discussion of publishing "positive" versus "negative" results; if only "positive" findings are published, this will result in a great deal of time and effort being wasted within the scientific community in repeating superficially appealing experiments. As such, transferring this knowledge to the public domain has value. The geoscientific modelling and measurement community (e.g. the volume of data generated by satellite retrievals) relies heavily on these data formats, and it is important that their refinement is an ongoing process.

Regardless of any ambiguity between the choice of bit-grooming or layer-packing, one clear result from this study is that simple linear packing typically results in *much* greater loss of precision than either of the two lossy methods discussed here. This is despite its widespread use.

Other useful contributions include the focus on the error-compression trade-off, the finding that the normalized entropy of the exponent field can be used to help determine which compression method is most appropriate, and the idea (introduced in the discussion) that the changes in the normalized entropy of the data could be used to determine how many significant figures should be retained.

**2 Reviewer 2**

**2.1 General comments**

1. *This paper describes a variant of lossy encoding which leverages the multi-dimensional nature of many scientific datasets that have greater data variances along different axes. The axes with small variations in data values are labeled "thin dimensions" and the axes with large variations in values are labeled "thick dimensions". The datasets are then "layer packed" with a linear scaling algorithm in the thin dimensions, recording a scale & offset value for each coordinate in the thick dimension.*
*I think the insights into the "thick" and "thin" dimensions are the primary value of this paper, with the actual compression algorithm and results being less important, overall.*

   Yes – one of the main things we are trying to do here is to assess whether treating different dimensions differently during gives much benefit over and above other methods that can be easily applied to such datasets. This is essentially trying to combine the best elements of GRIB and netCDF/HDF5.

**2.2 Specific comments**

1. *Applying the idea of thick & thin dimensions appropriately to other compression methods (such as the JPEG-2000 algorithm used in GRIB2) would be more valuable than just the idea of the simple scale & offset compression chosen.*

   We agree, and we spent a large amount of time trying to get this to work while preparing these revisions.
   In revising this work, we were able to run (after many technical hiccups) the same set of tests for GRIB/JPEG2000 compression as well (using 8, 12, 16 and 20 bits to represent the data). Our preliminary results showed that the JPEG2000 algorithm yields greater compression compared to the methods presented here for the same level of error; this echoes the findings of Caron (2014), which describe the efficient compression achieved. However like bit-grooming or layer-packing, JPEG2000 does not offer clear controls about the resultant errors and thus some experimentation (in setting the number of bits per value) is needed to avoid excessive loss of precision. We found that there was a large spread in the magnitude of the relative errors compared to the other methods considered. However the technical challenges required to convert a general netCDF field into GRIB format to be far in excess of what may be recommended to the average practitioner of geoscientific modelling. For this reason, and for the large spread in the compression and error result in the GRIB-compressed fields, and in order to keep the manuscript as focussed as possible, we chose not to include these results.

2. *Near the bottom of page 6, "for simplicity will have" should be corrected to "for simplicity we have".*

   Yes, well spotted. We have fixed this.

**2.3 Reviewer 2's comments to Review 1**

1. *Very nice review, much more detailed than mine. We seem to have homed in on the same insights: the differences in dimensions are the valuable part of the paper, and they aren't explored in enough detail to warrant a lot of enthusiasm in the current state of the*

*paper. My current feeling is a very "weak" accept, and I would prefer to ask for further exploration of the dimension ideas.*

Following the suggestions from Reviewer 1, the analysis and results have been considerably expanded and the revised manuscript offers further perspectives into the relationship between lossy compression, the resulting error and underlying complexity of the data.

**3 Reviewer 3**

**3.1 Summary**

*The paper introduces a "layer packing" lossy compression technique that takes advantage of the minimal horizontal variations in geoscience data relative to the larger variations across vertical dimensions. The layer packing technique is compared against many widely used lossless and lossy compression techniques and evaluated based on accuracy and time to solution. Layer packing is found to be beneficial in some cases while not in others, leading to the conclusion that care must be taken to evaluate whether lossy compression is worth the risk.*

**3.2 General Comments**

1. *The paper makes a good first attempt to evaluate the layer packing technique, but the paper would benefit from an additional revision. First, it's not clear what the paper is contributing. The authors state that the technique is used in GRIB (page 7, section 3) but that the evaluation was not possible due to relative error not being reported. Since the technique is not new, then the only contributions of the paper are the announcement of the general availability of the new non-GRIB tools, as well as the modestly detailed evaluation of the many compression techniques.*

The geoscientific modelling and remote-sensing community has to deal with the ever-growing volume of data generated. As such, it is important that the storage methods are reviewed in terms of the trade-off between compression, error and read/write times.
We have tried to avoid debate about data formats. Both have an important roles; the geoscientific community relies heavily on netCDF/HDF5, and GRIB remains the format of choice in many operational meteorological centres. Despite its excellent compression performance, GRIB can be regarded as less user-friendly.
The GRIB layer-packing is restricted to two-dimensional slices, whereas the layer-packing described here can operate on arbitrary hyperslices. The work presented in this manuscript aims to generate discussion about ways of incorporating the best of both methods.
With reference to the comment from Page 7, Section 3: "Caron (2014) estimated that GRIB2 files are on average 44% of the size of the equivalent deflate-compressed netCDF-4 files (n.b. relative errors were not reported, which limits the comparison)". The intended meaning was that the study of Caron (2014) reported the compression ratio, but not the relative errors, which makes it difficult to place the Caron (2014) results with those of this study.
The revisions to the original manuscript, focusing the analysis on the compressibility, errors and complexity of individual variables offers additional insights into these relationship and we believe adds substantially to the value of the paper.

**3.3   Specific Comments and Technical Corrections**

1. *The title, though catchy, is overloading the term "Goldilocks Zone" – the region around a star where perhaps liquid water might be found on a planet's surface. The title after the colon is clear on its own.*

   We have abbreviated the title, which as already been through several iterations.

2. *Page 2, line 3: "NetCDF" starts the last sentence on the line, though it should be "netCDF" for consistency.*

   We have revised for consistency of this term.

3. *Page 2, line 5: Why are three references necessary to describe the "deflate" compression method? Throughout the paper, be consistent with terms. scale-offset vs scale and offset. linear-packing vs linear packing.*

   Additional description of the deflate and shuffle algorithms has been added as suggested by Reviewer 1. We have reconsidered the references in this section. We have also reviewed the usage of the terms mentioned to improve the consistency of the manuscript.

4. *Page 3, line 30: I would suggest adding that ncdump is a command-line utility from the netCDF package because it might not be common knowledge. The paper introduces the "nc-packlayer" program and also uses other "nc"-prefixed tools from the NCO suite. For example, perhaps the following: "...(following the output format for the netCDF command-line utility ncdump)..."*

   Yes, this is correct, thanks for pointing this out. We have clarified this point.

5. *Page 3 (section 2 in general): More detail could be spent on the layer packing technique itself; the many monospaced examples of section 2 don't substantially add to the narrative and instead come across like a tutorial or README.*

   We have expanded the description of the algorithm itself. To keep the article short and concise, we have moved these details to an appendix.

6. *Page 4, line 11: run-on sentence*

   Thanks for pointing this out. This has been corrected.

7. *Page 4: The dollar symbol "$" is not explained, though I think you meant for it to refer to a shell variable syntax.*

   Yes, this is correct. This has been clarified.

8. *Page 5, Section 2.3: If I do the math correctly, the size of the datasets are (1) 962MB, (2) 267MB, (3) 68MB, (4) 613MB, (5) 30MB, and (6) 717MB. The rationale for the proposed compression is the growing volume of data in the geosciences, though none of these datasets are over a gigabyte in size. Compression of a multi-gigabyte dataset would make the argument more compelling, because datasets of such size will become more commonplace. Writing large datasets to disk as they are computed is a challenging problem and it would be nice to evaluate whether compressing large datasets is a viable option as they are generated. General comment about all Figures: Consider labeling the left and right panes of each figure as (a) and (b). For example, page 6, paragraphs starting on lines 9 and 17 sound too similar since Figure 1 is showing different things but is referred*

*to in the text in the same way. It would be more clear to say something like "Figure 1A shows..." and "Figure 1B presents..."*

The point about the magnitude of the file size is quite reasonable. We ran the test suite on variable of size 1.5 GB to examine the performance of the methods on larger datasets. This was included as an example referenced in the timing results, rather than adding it to the suite of variables presented in all results. This was mainly because, in the process of setting it up, the test suite was run many dozens of times; to accelerate the testing the variables considered were kept relatively small (the largest was about 65 MB).

However by the same token, the analysis for the revised manuscript has been done on individual variables alone, so the basic unit of study has become much smaller. While this might not impress those working with terabyte-scale data, it allows for greater insight into the methodology itself.

Regarding the figures, some of these have been moved to a supplementary material section. All panel plots now have labels (a), (b), etc., as suggested.

9. *Page 7: Starting on this page, for some reason all references to "figure 3" are lower case.*

   Thanks for pointing this out – it has been fixed.

10. *Page 8: Figure 1: The red and orange colors are too similar, though their position is clear from the legend.*

    All the figures have been thoroughly reworked. The color scheme in question no longer appears.

11. *Page 8, Figure 1, right panel: What does it mean to have the first column as "uncompressed" time since everything is normalized to DEFLATE? Was it the time to generate the data? Was it the time to copy the file?*

    Yes, in hindsight this wasn't very clear. It was effectively the time to copy the data. This bar is not included it in the revised manuscript. Thanks for drawing attention to it.

12. *Page 8, line 4: The reference to the HDF Group is used as an in-text citation as "(Group, 2016)". It would be best to fix your citation to not use HDF Group as a first/last name pair. See also your references on page 13, line 17.*

    Thanks for pointing this out. The default behaviour of the reference manager should have been over-ruled. This has been corrected.

13. *Page 9, line 1: run-on sentence*

    Thanks for pointing this out. It has been corrected.

14. *Page 10, Figure 3 caption: capitalize the Figure 1 and Figure 2 references.*

    This has been made more consistent.

15. *Page 11, line 6: misspelled "considered" – please consider a full spell check.*

    This has been fixed and we will ensure to run the spell checker again before resubmitting.

**References**

Caron, J. (2014). Converting GRIB to netCDF-4: Compression studies. `www.ecmwf.int/sites/default/files/elibrary/2014/13711-converting-grib-netcdf-4.pdf`, Last accessed: 2016-06-17. Presentation to the workshop "Closing the GRIB/netCDF gap", held at European Centre for Medium Range Weather Forecasts (ECMWF) at Reading, UK, 24-25 September 2014.

---

## Author Response (AR2)

**Author's response: "Finding the Goldilocks zone: Compression-error trade-off for large gridded datasets"**

Jeremy Silver, Charlie Zender

December 6, 2016

Dear GMD Editor,

Please find uploaded the final revised manuscript for manuscript GMD-2016-177. The only changes requested by the reviewers was capitalisation of one word in the Abstract. we have taken the opportunity to proof-read the manuscript and fixed a minor number of grammatical and formatting issues. The only change I would draw your attention to is a minor change to the title (now beginning with the word "The"). Appended to this document is the set of changes made to the text.

[revised manuscript text omitted]